# One-step construction of robust protocells and prototissues in water

Weixiao Feng[1,2], Peifan Li[1,2], Xin Li[1,2], Ziwei Wang[1,2], Min Chen[1,2], Yang Hu[1,3,4], Fu-Jian Xu [1,3,4] ✉ & Shaowei Shi [1,2] ✉

Bottom-up assembly of protocell building blocks into self-supporting, macroscopic, and robust prototissues that are stable in water and exhibit biomimetic behaviors remains a fundamental challenge. Here, we present a gas-liquid microfluidics-assisted diffusion-inhibited complexation strategy that enables low-cost, high-throughput fabrication of protocells and precise, large-scale (>10 cm) 3D construction of prototissues. The resulting prototissues display good mechanical integrity in both water and air and resist disintegration under external perturbations. By modulating the surrounding matrix or integrating bioactive components into the prototissue framework, programmable deformation/motion can be achieved through chemo-mechanical transduction. This strategy provides a platform for modeling the intricate behaviors of living systems and may facilitate applications in synthetic biology and bioengineering.

Protocells are miniature biomimetic systems that mimic the structure and function of natural cells, serving as simplified models for investigating complex physiological processes[1]. By integrating multiple protocells into interconnected 3D networks, hierarchical tissue-like structures, termed prototissues, can be further constructed, holding significant promise for bottom-up synthetic biology[2–4], tissue engineering[5], and bioinspired soft machines[6]. Various protocell models, including vesicles[7–10], proteinosomes[11–13], polymersomes[14], colloidosomes[15,16], coacervates[17–19], and microdroplets[20–22] have been explored for prototissue construction. In a series of pioneering studies, Bayley et al.[20] developed a strategy using droplet interface bilayers (DIBs) to create prototissues, with 3D printing facilitating spatial organization of lipid-coated water-in-oil droplets to generate millimeter-scale architectures with complex geometries. However, the DIB technique relies on an oil-phase medium, and the resulting prototissues exhibit poor stability under variations in pH, ionic strength, and temperature[2]. Recent advancements by Mann et al.[11–13] have introduced a water/oil Pickering emulsion approach, employing bioorthogonal chemistry to programmably assemble proteinosomes into prototissues. While this method offers advantages over lipid-

based systems, including enhanced protocellular adhesion, prolonged prototissue stability, and improved permeability, some drawbacks persist. For example, the synthetic chemistry may require harsh conditions or toxic reagents, the protocell organization lacks spatial control, and the removal of the oil phase complicates manufacturing processes. Currently, developing robust, self-supporting, and long-term stable protocells and prototissues in aqueous media, coupled with the achievement of scalability and low cost, has emerged as a critical challenge in this field[2,23].

In this work, we propose an electrostatically mediated diffusion-inhibited complexation (DIC) strategy to address this challenge by using negatively charged cellulose nanofibril (CNF) and a polycation (poly(diallyldimethylammonium chloride), PDDA) as building blocks to construct protocells and prototissues (Fig. 1). Gas-liquid microfluidic technology enables the efficient production of CNF/PDDA microcapsule-based protocells by spraying CNF-containing aqueous droplets into a PDDA-containing aqueous solution. During diffusion, CNF interacts electrostatically with PDDA, forming a robust and dense CNF/PDDA membrane that stabilizes the droplet, thereby generating protocells. CNF not only serves as a component of the membrane, but

[1]State Key Laboratory of Chemical Resource Engineering, Beijing University of Chemical Technology, Beijing, China. [2]Beijing Advanced Innovation Center for Soft Matter Science and Engineering, Beijing University of Chemical Technology, Beijing, China. [3]Key Laboratory of Biomedical Materials of Natural Macromolecules (Beijing University of Chemical Technology), Ministry of Education, Beijing, China. [4]Beijing Laboratory of Biomedical Materials, Beijing University of Chemical Technology, Beijing, China. ✉e-mail: xufj@mail.buct.edu.cn; shisw@mail.buct.edu.cn

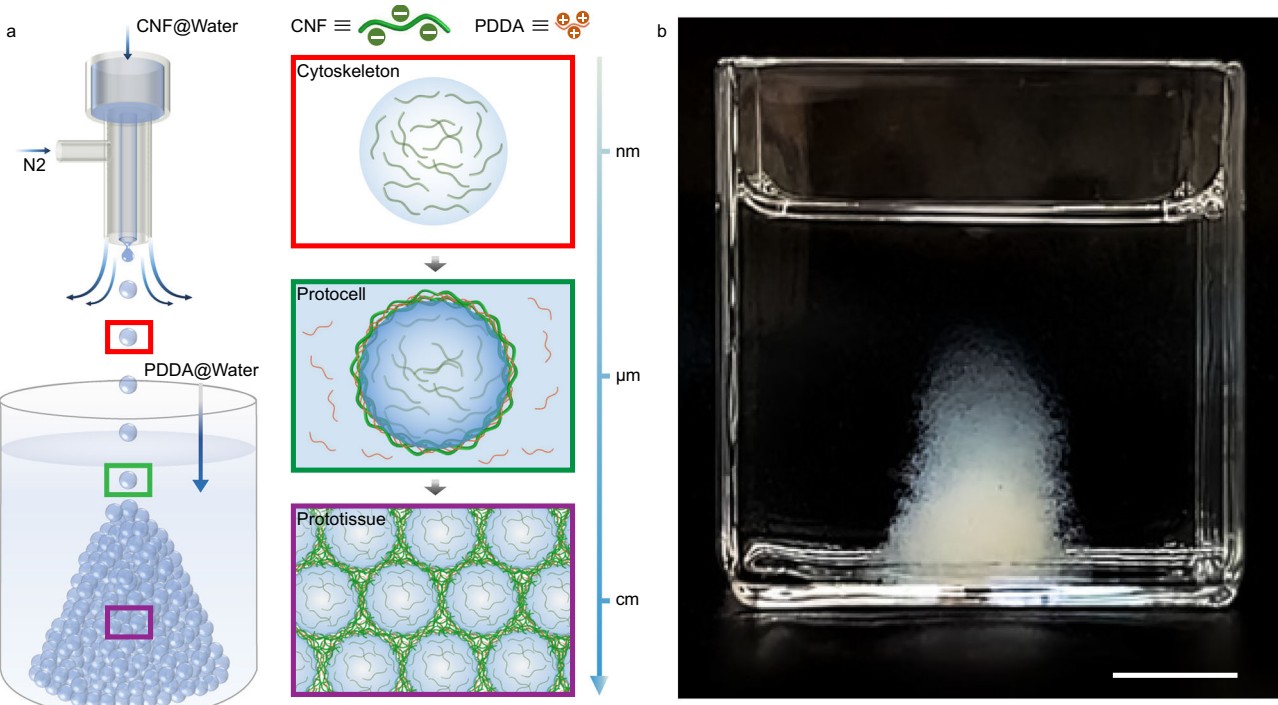

**Fig. 1 | Construction of protocells and prototissues. a** Schematic illustration of the one-step construction of protocells and prototissues via gas-liquid microfluidic-assisted DIC. **b** Photograph of a self-supporting prototissue in water. Scale bar, 1 cm.

also acts as a framework that supports the internal cavity of the protocells, similar to the cytoskeleton in natural cells, endowing the protocells with self-supporting properties. By integrating graphene oxide (GO) nanosheets into the membrane, the mechanical strength and permeability of protocells can be controlled, resulting in an enhanced elastic modulus and diverse diffusion behaviors of neutral polymers, polyanions, and polycations across the membrane. Importantly, the use of electrolytes to weaken the electrostatic interactions between CNF and PDDA induces adhesion between protocells, leading to the one-step construction of prototissues in aqueous media. Due to the robustness of the protocells and their strong inter-protocell adhesion, the resulting prototissues exhibit high structural stability and scalability, reaching sizes up to the decimeter scale. Finally, we demonstrate the precise 3D construction of prototissues via a molding strategy and highlight their chemo-mechanical transduction behaviors, such as osmotic pressure-induced deformation and buoyancy-driven motion.

## Results

### CNF/PDDA microcapsules for protocell modeling

By extruding an aqueous phase containing CNF (0.6 wt%) into an aqueous collection bath containing PDDA (0.5 wt%, $M_w$: 200-350 kDa) and Triton X-100 (0.1 wt%) at an extrusion rate of 30 mL h$^{-1}$ and a compressed gas flow rate of 1.0 L min$^{-1}$, CNF/PDDA microcapsules with diameters of ~ 800 μm were prepared at a rate of ~120 per minute (Fig. 2a, b, Supplementary Figs. 1, 2 and Supplementary Movie 1; the extruded aqueous phase was stained with amaranth to visualize microcapsule formation). Triton X-100 was used to reduce the air–water surface tension of the collection bath, thereby allowing droplets to pass through the surface and transform into microcapsules by DIC[24]. Without Triton X-100, or when its concentration was very low, the high interfacial tension prevented the formation of intact CNF/PDDA microcapsules (Supplementary Figs. 3, 4 and Supplementary Movie 2). In addition to Triton X-100, surfactants including Pluronic F68, Pluronic F127, Tween-20, and Solutol HS-15 also enabled

successful microcapsule formation (Supplementary Fig. 5). The produced CNF/PDDA microcapsules were washed with deionized water 10 times, then dispersed in deionized water for further experiments. The effects of concentration, pH, extrusion rate and gas flow rate on microcapsule formation and size were systematically investigated (Supplementary Figs. 6–8). CNF/PDDA microcapsules exhibited mechanical robustness and thermal stability, allowing them to withstand compression without rupture, remain self-supporting in air, and remain intact in boiling water for 30 min (Fig. 2c, Supplementary Figs. 9, 10). By mixing fluorescein isothiocyanate (FITC)-labeled CNF (CNF-FITC) into the droplets, microcapsules were visualized using confocal laser scanning microscopy (CLSM). 2D and reconstructed 3D fluorescence images revealed that CNF was distributed both on the membrane and inside the microcapsules, playing a role similar to that of the cytoskeleton, which is crucial for enhancing the mechanical properties of microcapsules (Fig. 2d–f). Using rhodamine B isothiocyanate (RITC)-labeled PDDA (PDDA-RITC) as a fluorescent probe, it was observed that PDDA was fully confined to the membrane and did not permeate into the interior (Supplementary Fig. 11). Scanning electron microscopy (SEM) characterization of freeze-dried CNF/PDDA microcapsules confirmed the dense CNF/PDDA membrane and 3D CNF network inside (Fig. 2g, h). Fluorescence recovery after photobleaching (FRAP) measurements showed that CNF had poor fluidity within the membrane and inside the microcapsules, as the fluorescence intensity exhibited almost no recovery after photobleaching in either region (Fig. 2i–k).

To assess the potential of CNF/PDDA microcapsules as protocell models, we investigated the membrane permeability. By dispersing CNF/PDDA microcapsules in hypertonic solutions of 1 M glucose or 1 wt% dextran of different molecular weights ranging from 10 to 500 kDa, no shrinkage of microcapsules was observed under an optical microscope (Fig. 2l, m and Supplementary Fig. 12). This indicated that the membrane of CNF/PDDA microcapsules did not selectively block glucose or dextran, and instead both solutes diffused into the microcapsules, leading to rapid equilibrium of osmotic pressure across the

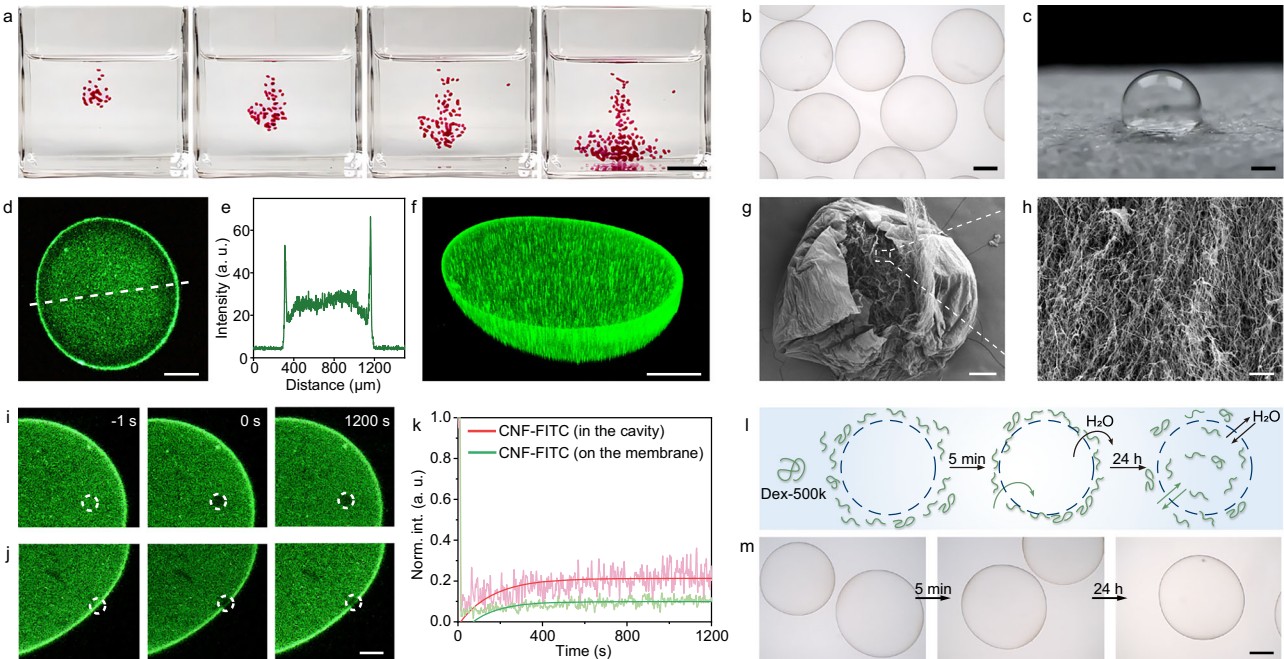

**Fig. 2 | Structure and permeability of CNF/PDDA microcapsules. a** Digital images showing the fabrication of CNF/PDDA microcapsules via gas-liquid microfluidic-assisted DIC, with the extruded phase stained with amaranth for enhanced visualization. Scale bar, 1 cm. **b** Optical microscopy image of CNF/PDDA microcapsules dispersed in water. Scale bar, 200 μm. **c** Digital image of a self-supporting CNF/PDDA microcapsule in air. Scale bar, 200 μm. 2D (**d**) and 3D reconstructed (**f**) CLSM images of a CNF/PDDA microcapsule containing CNF-FITC, and the corresponding fluorescence intensity profile (**e**, dashed line in (**d**)). Scale bar, 200 μm. SEM images of a ruptured, freeze-dried CNF/PDDA microcapsule, showing the membrane

membrane (**g**, scale bar, 100 μm) and the internal fibrillar network (**h**, scale bar, 10 μm). CLSM images showing the fluorescence recovery of CNF-FITC inside the microcapsule (**i**) and on the membrane (**j**) after photobleaching. Scale bar, 100 μm. **k** Fluorescence recovery curves of CNF-FITC inside the microcapsule and on the membrane. Schematic illustration (**l**) and optical microscopy images (**m**) showing the morphological evolution of CNF/PDDA microcapsules aged in a hypertonic solution of 1 wt% Dex-500k. Scale bar, 200 μm. Experiments in (**b–d**, **g**, **i**, **j**, **m**) were repeated independently 3 times with similar results. Source data are provided as a Source Data file.

membrane[25,26]. Using FITC-labeled dextran (Dex-FITC) as a fluorescent probe, CLSM imaging showed that Dex-FITC permeated across the membrane and diffused into the interior of microcapsules (Supplementary Fig. 13). In addition to neutral hypertonic solutions of glucose or dextran, microcapsules were also dispersed in a polyanion aqueous solution containing 1 wt% poly(acrylic acid) (PAA-2k, $M_w$: ~2 kDa). Due to the electrostatic repulsion between PAA-2k and CNF, PAA-2k could not diffuse into the microcapsules, while the water inside the microcapsules could diffuse out, leading to microcapsule shrinkage (Supplementary Fig. 14). When the microcapsules were redispersed in water, they returned to a spherical shape to balance the osmotic pressure. These results demonstrate the elastic nature of the membrane. Consequently, regulating the permeability of CNF/PDDA microcapsules was key to constructing protocells for further biomimetic applications.

## CNF-GO/PDDA microcapsules for protocell modeling

To address the above issue, negatively charged GO nanosheets were incorporated into the CNF/PDDA membrane due to their large aspect ratio and specific surface area[27,28]. The addition of 0.2 wt% GO into the extruded aqueous phase did not interfere with the preparation of microcapsules and produced CNF-GO/PDDA microcapsules with a narrow size distribution, mechanical robustness, and biocompatibility (Supplementary Figs. 15–17, and Supplementary Movie 3). Using CNF-FITC and pyrene-labeled GO (GO-Pyrene) as fluorescent probes (Supplementary Figs. 18–20), CLSM images showed that CNF was distributed on the membrane and in the cavity, similar to the case of CNF/PDDA microcapsules (Figs. 3a–c and 2d–f). GO, on the other hand, was mainly distributed on the membrane, as the fluorescent signal inside the microcapsule was weak (Fig. 3c). This behavior could be attributed to the amphiphilic nature of GO, which possessed a hydrophilic surface

and a hydrophobic basal plane[29,30], making it more likely to accumulate at the air–water interface during gas shearing and form a CNF-GO/PDDA membrane by DIC. It was also difficult to observe GO nanosheets within CNF networks inside the microcapsules by SEM (Fig. 3d, e).

CNF-GO/PDDA microcapsules exhibited completely different permeability behavior compared to CNF/PDDA microcapsules. In a hypertonic solution of 1 M glucose, CNF-GO/PDDA microcapsules first shrank due to the difference in osmotic pressure, then gradually returned to their original shape (spherical) within 60 min as a result of the increased concentration of glucose and osmotic pressure inside the microcapsules (Fig. 3f, g). By analyzing the spherical microcapsule percentage (SMP, defined as the ratio of the number of spherical microcapsules to the total number of microcapsules) over time, a sigmoidal curve was obtained that reflects the kinetics of glucose diffusion (Fig. 3h)[31–33]. The diffusion rate of glucose could be well controlled by varying the concentration of GO in the extrusion droplet. The lower the concentration of GO, the faster the diffusion rate. These results demonstrated that the incorporation of GO significantly reduced the porosity of the membrane (in the following discussions, the GO concentration in the extrusion droplets was fixed at 0.2 wt% unless otherwise noted). When dispersing microcapsules in a hypertonic solution of dextran, a much longer time was required for the transmembrane diffusion of dextran, due to its larger hydrodynamic volume. Taking dextran with molecular weights of 100 and 500 kDa (termed Dex-100k and Dex-500k) as examples: for Dex-100k, the SMP reached 100% after 20 h, whereas for Dex-500k, the SMP only reached 40% after 40 h (Fig. 3i and Supplementary Figs. 21, 22). By encapsulating FITC-labeled Dex-100k (Dex-100k-FITC) and rhodamine B isothiocyanate (RITC)-labeled Dex-500k (Dex-500k-RITC) in CNF-GO/PDDA microcapsules and aging for 8 h in water, CLSM was used to probe the distribution of dextran. The fluorescence intensity of Dex-

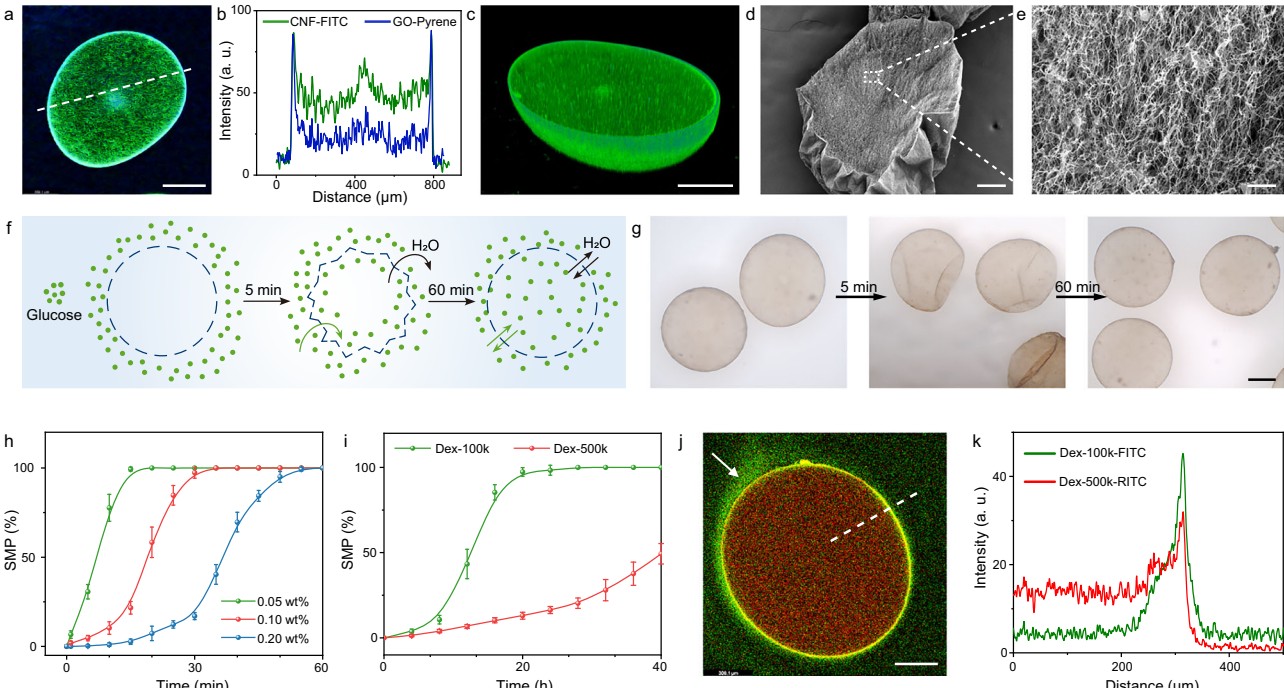

**Fig. 3 | Structure and permeability of CNF-GO/PDDA microcapsules.** 2D (**a**) and 3D reconstructed (**c**) CLSM images of a CNF-GO/PDDA microcapsule containing CNF-FITC and GO-Pyrene, and the corresponding fluorescence intensity profiles (**b**, dashed line in **a**). Scale bar, 200 μm. SEM images of a ruptured, freeze-dried CNF-GO/PDDA microcapsule, showing the membrane (**d**, scale bar, 100 μm) and the internal framework (**e**, scale bar, 10 μm). Schematic illustration (**f**) and optical microscopy images (**g**) showing the morphological evolution of CNF-GO/PDDA microcapsules aged in a hypertonic solution of 1 M glucose. Scale bar, 200 μm.

Time-dependent evolution of SMP in hypertonic solutions of 1 M glucose at varying GO concentrations (**h**), and SMP in hypertonic solutions of 1 wt% Dex-100k and Dex-500k (**i**). Data are presented as mean values ± standard deviation (*n* = 3 independent experiments). CLSM image (**j**) showing a CNF-GO/PDDA microcapsule encapsulating Dex-100k-FITC and Dex-500k-RITC, and the corresponding fluorescence intensity profiles (**k**, dashed line in (**j**)). Scale bar, 200 μm. Experiments in (**b**, **d**, **g**–**j**) were repeated independently 3 times with similar results. Source data are provided as a Source Data file.

100k was similar outside and inside the microcapsule, while the fluorescence intensity of Dex-500k inside the microcapsule was stronger than outside, confirming the selective permeability of CNF-GO/PDDA membrane (Fig. 3j, k).

In addition to neutral glucose and dextran, we further investigated the effects of hypertonic solutions of polyanions and polycations on CNF-GO/PDDA microcapsules. For the polyanion PAA-2k, CNF-GO/PDDA microcapsules exhibited behavior similar to that of CNF/PDDA microcapsules but with reduced collapse, indicating enhanced mechanical strength of CNF-GO/PDDA microcapsules due to the incorporation of GO nanosheets (Fig. 4a, b and Supplementary Fig. 14). By dispersing CNF-GO/PDDA microcapsules in an aqueous solution of the polycation PDDA (PDDA-100k, $M_w$: 100-200 kDa), shrinkage of microcapsules was observed. However, when redispersed in water, the microcapsules remained shriveled and lost the ability to undergo elastic recovery (Fig. 4c, d). This could be attributed to the transmembrane diffusion of PDDA-100k, which crosslinked the CNF/GO inside the microcapsules, thereby inhibiting the swelling of microcapsules induced by osmotic pressure. A control experiment using higher molecular weight PDDA (PDDA-400k, $M_w$: 400-500 kDa) showed typical shrink-swell behavior. Due to its larger hydrodynamic volume, PDDA-400k was unable to penetrate the membrane to crosslink the interior of the microcapsules, allowing microcapsules to swell upon redispersion in water. Notably, compared to their initial state, the surface of microcapsules produced by secondary swelling exhibited discontinuous dark microdomains, suggesting a certain degree of cross-linking between PDDA-400k and the membrane (Fig. 4e, f and Supplementary Fig. 23). When polyethyleneimine with a very low molecular weight of 0.6 kDa (PEI-0.6k) was used, an interesting aggregation behavior was observed inside the microcapsules

(Fig. 4g, h and Supplementary Fig. 24). PEI-0.6k was able to diffuse into the microcapsules and electrostatically interact with CNF/GO. However, in contrast to PDDA-100k, which bore permanently charged quaternary ammonium groups, the protonatable amine groups along the PEI-0.6k chains provided weaker electrostatic interactions. In addition, the much lower molecular weight and shorter chain length of PEI-0.6k limited the spatial extent of electrostatic crosslinking with CNF/GO, resulting in a more localized and mechanically weaker network. Consequently, the PEI-0.6k-induced crosslinked structure was unable to resist osmotic swelling upon redispersion of microcapsules in pure water, leading to network disruption, fragmentation, and the formation of small fragments that gradually associated into larger aggregates. Using FITC-labeled PEI-0.6k as a fluorescent probe and characterizing the aggregates by FRAP, we observed almost no fluorescence recovery after photobleaching, indicating the limited mobility of PEI (Supplementary Fig. 25).

## Chemical communication in CNF-GO/PDDA protocells

Having established the membrane permeability of CNF-GO/PDDA microcapsules, we employed an enzymatic cascade to demonstrate the biomimetic potential of this protocell system. Two enzymes, horseradish peroxidase (HRP) and glucose oxidase (GOx), were incorporated into the CNF-GO/PDDA protocells by mixing them into the extrusion droplets. Due to the negative charge of both HRP and GOx under neutral conditions, they participated in membrane formation via DIC, as evidenced by the strong fluorescence signals of FITC-labeled HRP and RITC-labeled GOx on membrane (Fig. 5a, b). HRP and GOx were also distributed inside the protocells, with a lower content of HRP detected. Compared to GOx, HRP contained more hydrophobic groups, enhancing its interfacial activity at the air-water

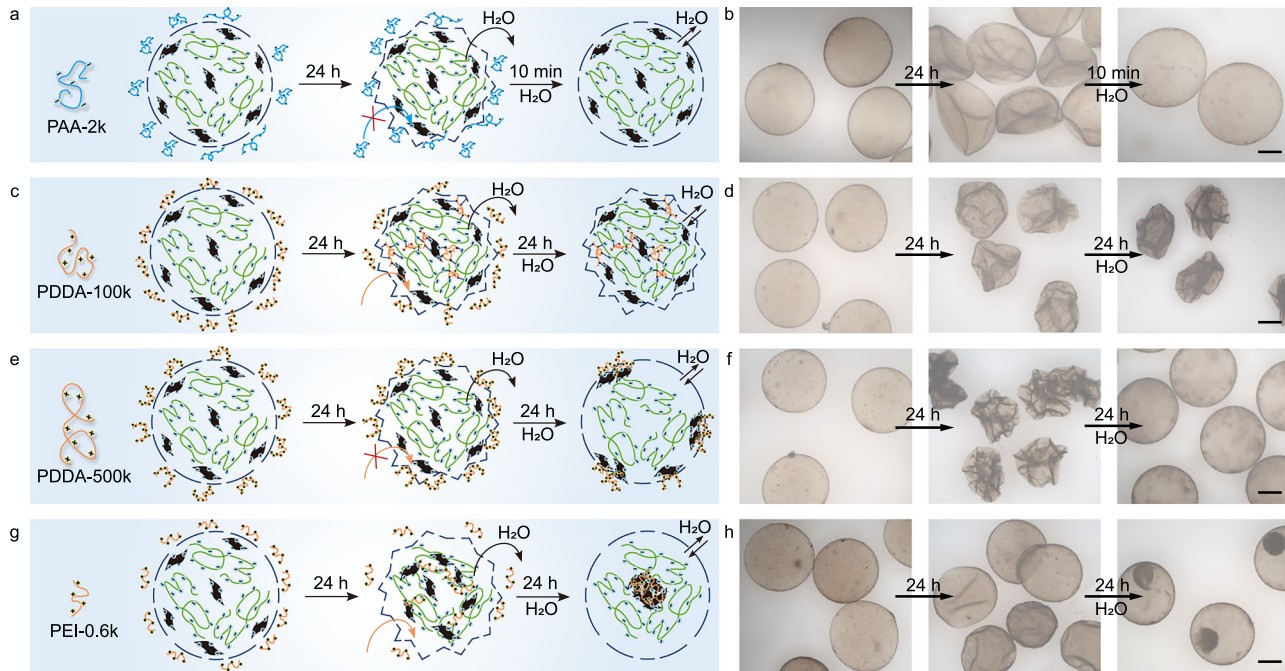

**Fig. 4 | Osmotic responsiveness of CNF-GO/PDDA microcapsules. a–h** Schematic illustrations and optical microscopy images showing the morphological evolution of CNF-GO/PDDA microcapsules first aged in different hypertonic solutions and subsequently in pure water: 1 wt% PAA-2k (**a**, **b**), 1 wt% PDDA-100k (**c**, **d**), 1 wt% PDDA-400k (**e**, **f**), and 1 wt% PEI-0.6k (**g**, **h**). Scale bar, 200 µm. Experiments in (**b**, **d**, **f**, **h**) were repeated independently 3 times with similar results.

interface and facilitating the DIC process, similar to the case of GO-containing protocells[34]. CNF-GO/PDDA protocells effectively retained HRP and GOx, with no leakage observed for 5 days (Supplementary Fig. 26). The enzymatic cascade was triggered by the addition of glucose and benzoyl leuco methylene blue (BLMB) to the continuous phase (Fig. 5c). Glucose diffused into the protocells and was catalyzed by GOx to release hydrogen peroxide ($H_2O_2$) first; subsequently, the non-fluorescent BLMB that had diffused into the protocells was oxidized by $H_2O_2$ under HRP catalysis, producing fluorescent methylene blue (MB). CLSM imaging showed that MB initially concentrated near the membrane due to enzyme enrichment at membrane, and gradually diffused into the interior and exterior of the protocells over time (Fig. 5d, e). To investigate the chemical signaling between protocells, HRP and GOx were separately loaded in two populations of CNF-GO/PDDA protocells. In the cascade reactions, $H_2O_2$ was produced by GOx-containing protocells, then diffused into HRP-containing protocells, where it triggered HRP-catalyzed oxidation of BLMB to MB (Supplementary Fig. 27a). Tracking the fluorescence signal of MB showed that MB first appeared in the region of HRP-containing protocells, before gradually diffusing to the region of GOx-containing protocells (Supplementary Fig. 27b). In comparison to protocells co-encapsulating HRP and GOx, the reaction kinetics were slower due to the longer diffusion time of chemicals between different protocells.

## One-step construction of 3D prototissues by protocells

The robustness and oil-free production of CNF-GO/PDDA protocells enabled the construction of prototissues. Here, we propose a straightforward strategy to achieve the assembly of protocells by adding 0.3 M NaCl into the extruded aqueous phase. As the droplets entered the collection bath to form CNF-GO/PDDA protocells, they were not dispersed in water but instead adhered to each other, creating a self-supporting tissue-like structure at the centimeter scale, which could be further transferred to air. The protocells exhibited sufficiently strong adhesion, enabling the prototissue to withstand needle-induced disturbances without disintegration (Fig. 6a and Supplementary Movie 4). Freeze-drying the CNF-GO/PDDA prototissue

yielded a fully dehydrated, tissue-like material that retained its structural integrity and was capable of supporting external loads (Supplementary Fig. 28). The constructed conical prototissue could be scaled up to a height of >6.5 cm and a diameter of >5 cm, comprising >20,000 protocells (Fig. 6b).

The primary mechanism underlying microcapsule-microcapsule adhesion should be attributed to the electrolyte-induced screening of electrostatic repulsion[35]. Without electrolyte, CNF/GO and PDDA assembled into well-stratified structures via strong electrostatic interactions, with a predominantly positively charged outer surface and a negatively charged interior. This charge asymmetry led to strong electrostatic repulsion between individual microcapsules, preventing adhesion. Upon addition of electrolyte, increased ionic strength weakened the electrostatic complexation between CNF/GO and PDDA, thereby enhancing the mobility of components within the membrane and leading to the formation of a more homogeneous membrane. This membrane contained a uniform distribution of positive and negative charges, enabling microcapsules to adhere upon contact. Meanwhile, the weakened electrostatic interactions also promoted interdiffusion between CNF/GO and PDDA during complexation, which may lead to membrane thickening. Using CNF-FITC as a fluorescent probe, CLSM imaging visualized the thickened CNF-GO/PDDA membranes and protocell adhesion (Fig. 6d and Supplementary Fig. 29). SEM analysis of a freeze-dried prototissue demonstrated that the protocells were tightly interconnected via CNF/GO networks (Supplementary Fig. 30).

To further clarify the role of ionic strength in microcapsule adhesion, CNF-GO/PDDA prototissues were prepared at different NaCl concentrations (Supplementary Fig. 31). At low NaCl concentrations (0.1 M), relatively strong electrostatic repulsion prevented microcapsule adhesion and prototissue formation. Stable prototissues were formed at intermediate NaCl concentrations (0.2 and 0.3 M), where effective microcapsule adhesion occurred. At higher NaCl concentrations (0.4 and 0.5 M), electrostatic interactions between CNF/GO and PDDA were excessively weakened, leading to extensive interdiffusion between the CNF/GO-containing extruded aqueous phase and the PDDA-containing collection bath, thereby inhibiting the formation of

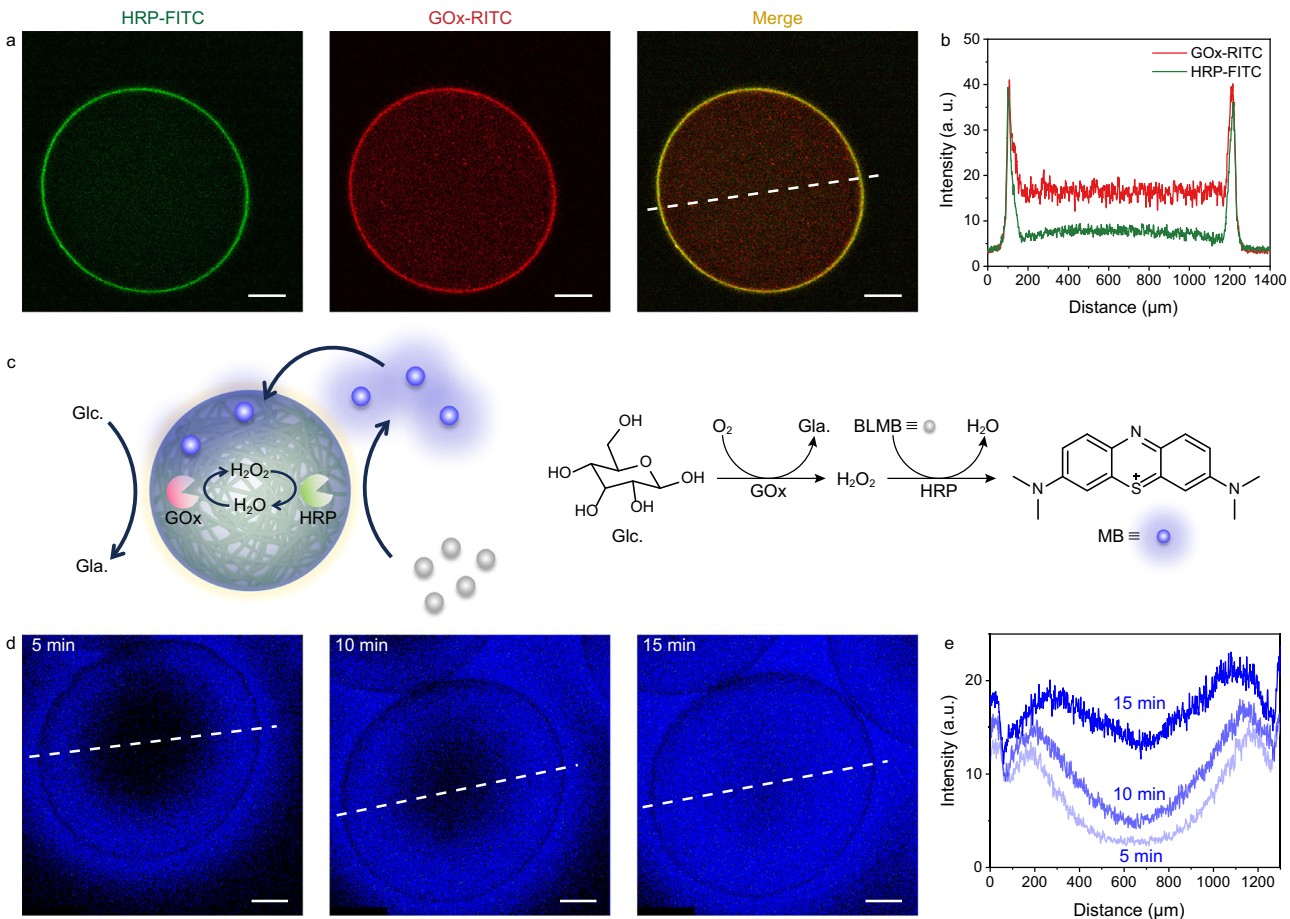

**Fig. 5 | Enzymatic reactions in protocells. a** CLSM images of CNF-GO/PDDA microcapsule loaded with HRP-FITC and GOx-RITC. Scale bar, 200 μm. **b** Fluorescence intensity profiles along the white dashed line in (**a**). **c** Schematic illustration of the enzymatic cascade reaction occurring within the protocell.

**d** CLSM images showing the spatial distribution of MB molecules at different times. **e** Fluorescence intensity profiles at different time along the white dashed line in (**d**). Experiments in a and d were repeated independently 3 times with similar results. Source data are provided as a Source Data file.

stable complex membranes and microcapsules. In control experiments, 2D macroscopic CNF-GO/PDDA membranes were prepared under different NaCl concentrations and characterized by atomic force microscopy (AFM). The results showed that the membrane thickness increased with higher NaCl concentrations, confirming the proposed mechanism (Supplementary Fig. 32). Beyond NaCl, other electrolytes, including KCl, CaCl$_2$, MgCl$_2$, Na$_2$SO$_4$, as well as neutral or basic buffer systems were also effective in regulating protocell adhesion (Supplementary Fig. 33).

Having achieved the stable construction of macroscopic prototissues, we investigated their osmotic responsiveness in polyanion solutions to determine whether the behavior of individual protocells could be integrated into the prototissues. By immersing the conical prototissue composed of CNF-GO/PDDA protocells into a polyanion solution of 1 wt% PAA-2k, a pronounced volume shrinkage was observed within 30 min due to water loss from the individual protocells (volume shrinkage ratio: ~45%, calculated as $1 - V/V_0$ based on cone volume before and after shrinkage). Upon replacing the PAA-2k solution with deionized water, the prototissue reabsorbed water and recovered to its original volume within 30 min as a result of decreased osmotic pressure (Fig. 6e and Supplementary Movie 5). When the concentration of PAA-2k was further increased to 3 wt% and 5 wt%, the elevated osmotic pressure led to more significant water loss from the prototissue, with shrinkage ratios reaching 60% and 80%, respectively. Owing to the robust nature of CNF-GO/PDDA protocells, the

prototissue was able to undergo multiple shrinkage–swelling cycles without rupture or disintegration (Fig. 6f).

The integration of gas-liquid microfluidic technology with molding enabled the precise 3D construction of prototissues. Patterned molds were pre-positioned in the collection bath, and protocells were deposited into the molds through manual control of the needle during the gas-spraying process (Fig. 6g and Supplementary Movie 6). After demolding, prototissues with well-defined shapes such as humanoid, cloverleaf, and Mickey Mouse-like geometries were readily obtained. The molded prototissues remained intact under vigorous agitation, further confirming the strong inter-protocell adhesion between protocells (Supplementary Fig. 34 and Supplementary Movie 7). Moreover, by alternately stacking CNF-GO/PDDA and CNF/PDDA protocells, a Janus-type starfish-shaped prototissue was fabricated (Fig. 6i). When placed in a 1 wt% PDDA-100k aqueous solution, both layers of the prototissue experienced water loss and shrinkage. However, due to the incorporation of GO nanosheets, the bottom layer composed of CNF-GO/PDDA protocells exhibited higher mechanical strength and resisted deformation under osmotic pressure. In contrast, the top layer composed of CNF/PDDA protocells, with a softer membrane structure, collapsed more significantly upon dehydration (Supplementary Fig. 35). Owing to their different mechanical properties and osmotic responsiveness, the two layers underwent unequal magnitudes and rates of volume contraction under the same hypertonic conditions, thereby driving

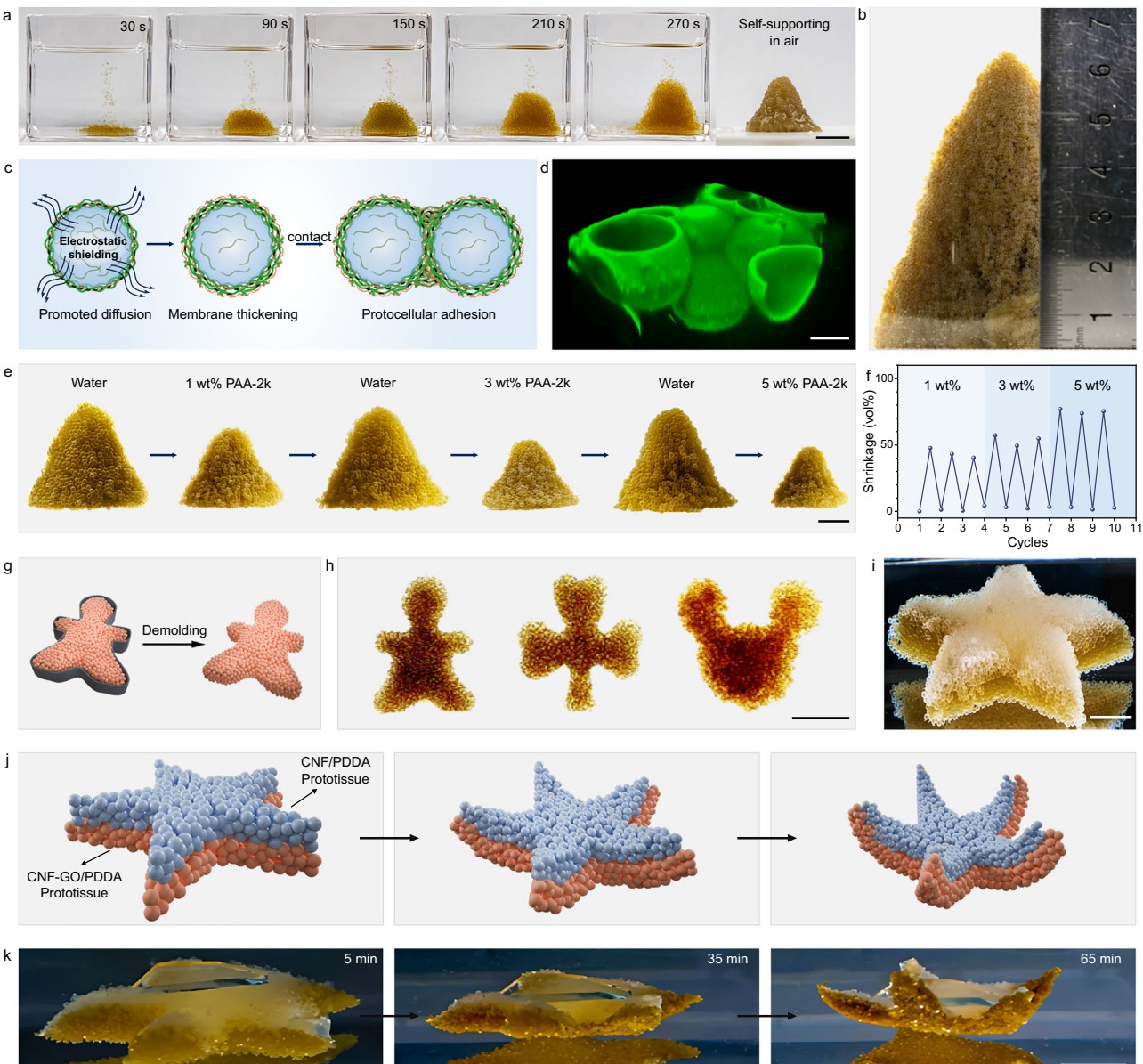

**Fig. 6 | Construction of prototissues and osmotic pressure-induced deformation. a** Digital images showing the construction of a CNF-GO/PDDA prototissue via gas-liquid microfluidic-assisted DIC and its self-supporting properties in water and air. Scale bar, 1 cm. **b** Digital image showing a large-scale, conical CNF-GO/PDDA prototissue in water (height > 6.5 cm, diameter > 5 cm). **c** Schematic illustration of the electrolyte-mediated protocellular adhesion. **d** 3D reconstructed CLSM image showing the structure of CNF-GO/PDDA prototissue containing CNF-FITC. Scale bar, 200 μm. Digital images (**e**) showing the osmotic pressure-induced shrinkage and swelling of a CNF-GO/PDDA prototissue in different aqueous environments, and quantitative analysis (**f**) of volume changes over multiple shrinkage–swelling cycles. Scale bar, 0.5 cm. **g** Schematic illustration of the prototissue construction with defined geometry via a molding strategy. **h** Digital images showing molded prototissues with humanoid, cloverleaf, and Mickey Mouse-like geometries. Scale bar, 2 cm. **i** Digital image of a Janus starfish-like prototissue composed of CNF-GO/PDDA protocells (bottom layer) and CNF/PDDA protocells (top layer). Scale bar, 1 cm. Schematic illustration (**j**) and digital images (**k**) showing the osmotic pressure-induced deformation of a Janus starfish-like prototissue in a hypertonic solution of 1 wt% PDDA-100k. A glass slide was placed on top of the prototissue to suppress buoyancy caused by variations in the surrounding solution density. Scale bar, 1 cm. Source data are provided as a Source Data file.

the bending of bilayer structure (Fig. 6j, k and Supplementary Movie 8).

**Programmable chemo-mechanical transduction of prototissues**
By mixing 0.02 wt% catalase into the extruded aqueous phase to generate CNF-GO/PDDA protocells, we employed enzymatic reactions to demonstrate the lifelike potential of protocells and prototissues[33]. For individual protocells, CLSM imaging confirmed that FITC-labeled catalase was distributed both on the membrane and inside the protocells, suggesting its involvement in membrane formation via DIC,

owing to its negative charge under neutral conditions (Fig. 7a). Upon the addition of hydrogen peroxide ($H_2O_2$, 0.3 wt%) to the surrounding water, oxygen bubbles were generated inside the protocells and as oxygen accumulated, the resulting buoyant force overcame gravity, causing the protocells to rise (Fig. 7b, c and Supplementary Movie 9). Prototissues exhibited similar buoyancy-driven motion, which could be harnessed for chemo-mechanical transduction (Fig. 7d and Supplementary Movie 10). By varying the concentration of $H_2O_2$ or catalase, the buoyancy velocity of prototissues could be tuned (Supplementary Fig. 36).

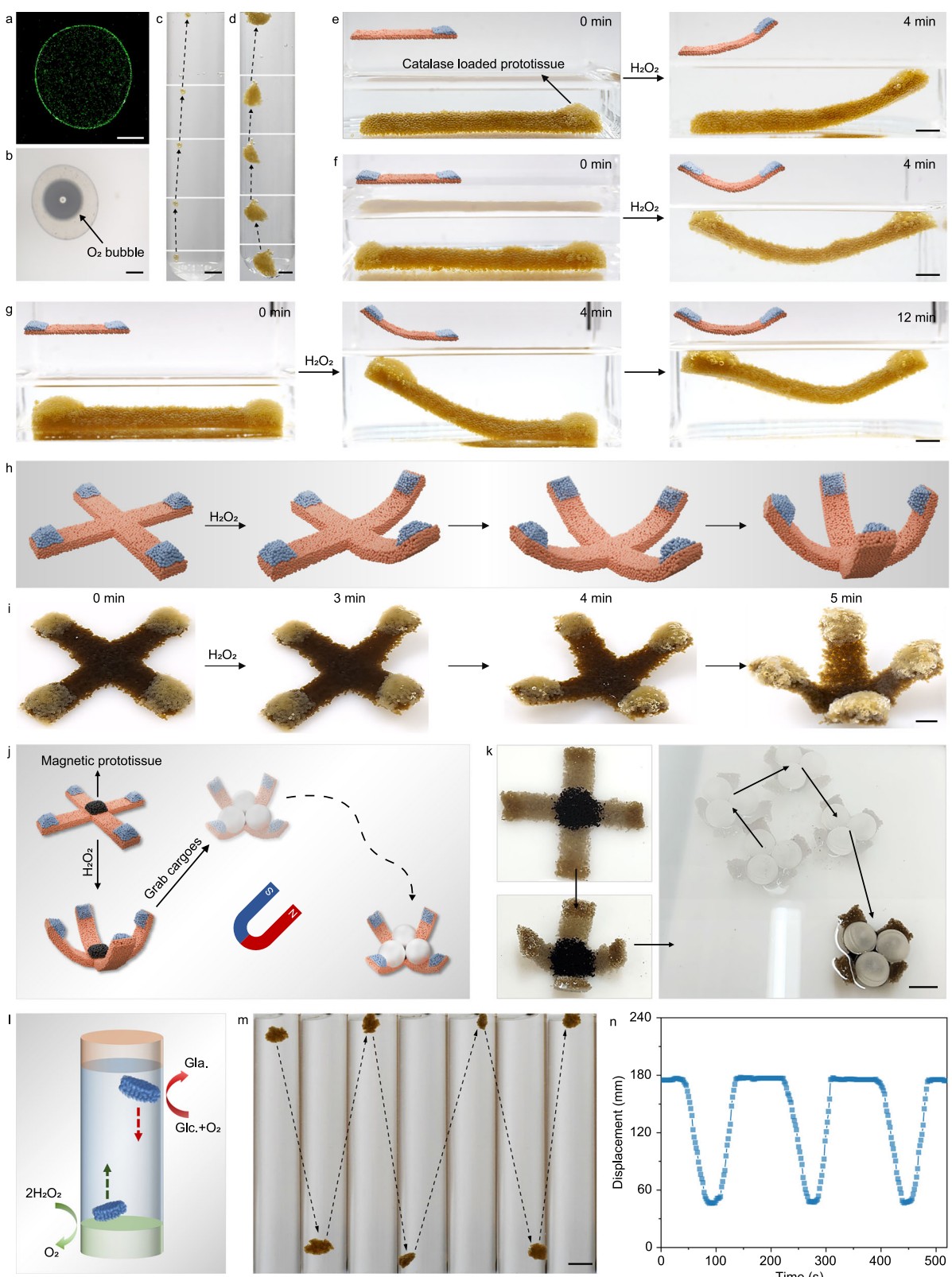

By assembling catalase-loaded and unloaded CNF-GO/PDDA protocells into a single prototissue, we investigated the programmable chemo-mechanical transduction and complex motion behavior[36–38]. A strip-shaped, unloaded prototissue was first fabricated via molding. When catalase-loaded protocells were deposited at one end, the subsequent addition of $H_2O_2$ to the surrounding water led to localized oxygen generation, resulting in asymmetric buoyancy and upward bending of the prototissue (Fig. 7e and Supplementary Movie 11). When catalase-loaded protocells were deposited at both ends or in the middle of the prototissue, different bending behaviors were observed, forming either concave or convex shapes depending on the location of buoyancy (Fig. 7f, Supplementary Fig. 37 and Supplementary Movie 12,

**Fig. 7 | Buoyancy-driven motion/deformation of prototissues. a** CLSM image of a CNF-GO/PDDA protocell encapsulating FITC-labeled catalase. Scale bar, 200 µm. **b** Optical microscopy image of a catalase-loaded CNF-GO/PDDA protocell encapsulating an oxygen bubble. Scale bar, 200 µm. Digital images showing the buoyancy-driven motion of catalase-loaded CNF-GO/PDDA protocells (**c**) and prototissues (**d**) upon the addition of $H_2O_2$ into the surrounding aqueous phase. Scale bar, 5 mm. **e** Schematic illustration and digital images showing the asymmetric upward bending of a strip-shaped prototissue with catalase-loaded CNF-GO/PDDA protocells deposited at one end. Scale bar, 1 cm. **f** Schematic illustration and digital images showing the symmetric upward bending of a strip-shaped prototissue with catalase-loaded CNF-GO/PDDA protocells deposited at both ends. Scale bar, 1 cm.

**g** Schematic illustration and digital images showing the time-dependent upward bending of a strip-shaped prototissue induced by depositing CNF-GO/PDDA protocells with different catalase concentrations at both ends. Scale bar, 1 cm. Schematic illustration (**h**) and digital images (**i**) showing the upward bending of a cross-shaped prototissue with catalase-loaded CNF-GO/PDDA protocells at four ends. Scale bar, 1 cm. Schematic illustration (**j**) and digital images (**k**) showing the cross-shaped prototissue functioning as a soft gripper. Scale bar, 2 cm. Schematic illustration (**l**) and digital images (**m**) showing the vertical oscillatory motion and displacement-time trace (**n**) of CNF-GO/PDDA prototissue. Scale bar, 1 cm. Experiments in a and b were repeated independently 3 times with similar results. Source data are provided as a Source Data file.

13). Furthermore, by tuning the catalase concentrations in the protocells at both ends (0.2 wt% on the left and 0.1 wt% on the right), 4D motion/deformation could be achieved, where asymmetric reaction rates caused one end to rise earlier than the other (Fig. 7g and Supplementary Movie 14). A cross-shaped prototissue was also fabricated, with catalase-loaded protocells deposited at its four ends. Upon exposure to $H_2O_2$, buoyancy induced upward bending of the four arms, transforming the prototissue into a cage-like structure capable of capturing hollow plastic spheres (Supplementary Movie 15). By further incorporating protocells containing NdFeB ferromagnetic microparticles into the central region, the resulting soft gripper could be directed to translate and rotate under an external magnetic field (Fig. 7h–k and Supplementary Movie 16).

Finally, we investigated a buoyancy-driven oscillatory system. A small CNF–GO/PDDA prototissue encapsulating 0.1 wt% catalase and 0.1 wt% GOx was prepared and placed in a water-filled glass tube (diameter = 2 cm, Fig. 7l). An agarose hydrogel containing 3 wt% $H_2O_2$ was fixed at the bottom of tube, while an agarose hydrogel containing 18 wt% glucose was fixed at the top. $H_2O_2$ and glucose gradually diffused from the hydrogels into the surrounding water. When the prototissue approached the bottom $H_2O_2$ reservoir, catalase decomposed $H_2O_2$ into $O_2$, generating gas bubbles inside the protocells and increasing buoyancy, which caused the prototissue to rise. Upon reaching the top glucose reservoir, GOx consumed the encapsulated $O_2$, leading to a decrease in buoyancy and causing the prototissue to sink back toward the bottom. In this manner, the prototissue underwent repeated up-down motion between the two agarose hydrogels until the $H_2O_2$ and glucose were largely depleted, after which the prototissue gradually ceased motion and settled at the bottom. Under the experimental conditions used, the prototissue exhibited three reproducible oscillation cycles (Fig. 7m, n and Supplementary Movie 17).

## Discussion

Engineering functional prototissues is a compelling frontier in life science, aiming at replicating the structural and dynamic complexity of living systems. We demonstrate a one-step aqueous strategy for constructing protocells and prototissues via DIC between CNF and polycations. This scalable and efficient approach yields self-supporting prototissues with high mechanical integrity, offering a robust alternative to existing emulsion droplet- and coacervate droplet-based systems. By modulating osmotic pressure or utilizing buoyancy and magnetic fields, diverse deformation and motion behaviors can be achieved, offering great potential for applications in soft robotics and biomedical devices. These prototissues hold promise for future use as scaffolds in tissue engineering, supporting cell encapsulation and activity for tissue repair and regeneration. Despite these advances, several aspects remain to be explored. For example, further tuning of membrane composition and/or post-modification of the membrane could enable more precise control over permeability and a tighter molecular-weight cut-off, which are important for broadening the range of encapsulation. Integrating

additional actuation mechanisms (e.g., light and temperature) is expected to enable more complex and programmable motion and deformation of prototissues.

## Methods

### Materials

All materials were used without further purification. Cellulose nanofibril (CNF) aqueous dispersion (1.0 wt%, from cotton) was purchased from Guilin Qihong Tech. Ltd. Poly(diallyldimethylammonium chloride) (PDDA, 20 wt% aqueous solution, $M_w$: 100–200 kDa, 200–350 kDa, 400–500 kDa) was purchased from Shanghai Aladdin Biochemical Technology Co., Ltd. The hydrodynamic sizes of three PDDA samples, measured by dynamic light scattering in dilute aqueous solution, are ~23.4, 29.4, and 40.5 nm, respectively. Triton X-100 (BR), hydrogen peroxide ($H_2O_2$) (30 wt% aqueous solution), N-(1-Pyrenyl) maleimide (≥98%), thionyl chloride (≥99%), L-Cysteine (≥98%), triethylamine (≥98%), and polyethyleneimine ($M_w$: 600 Da) were purchased from Aladdin. Fluorescein 5(6)-isothiocyanate (FITC, ≥96%, mixture of 5- and 6- isomers), rhodamine B isothiocyanate (RITC, ≥ 96%, mixture of isomers), dextran ($M_w$: 10 kDa, 40 kDa, 70 kDa, 100 kDa, 500 kDa), tris(2-carboxyethyl)phosphine (TCEP, ≥98%), and poly(acrylic acid) (PAA, 40 wt% aqueous solution, $M_w$: 2 kDa) were purchased from J&K. β-D-glucose (≥99%) was purchased from TCI. Graphene oxide (GO) was purchased from XFNANO. Catalase (1000 U mg$^{-1}$), carbonate buffer (0.1 M, pH 9.2), PBS buffer (0.1 M, pH 7.4), and agarose (≥99%) were purchased from Shanghai Yuanye. Sodium chloride (≥99.5%), potassium chloride (≥99.5%), calcium chloride (≥96%), magnesium chloride (≥98%), and anhydrous sodium sulfate (≥99%) were purchased from Fuchen Chemical. Milli-Q water (18.2 MΩ•cm) was used in the preparation of all aqueous solutions.

### Characterizations

The preparation, self-supporting, and chemo-mechanical transduction behaviors of protocells and prototissues were recorded as images or movies with a digital camera (SONY Alpha 6400 equipped with Tamron 17–70 mm F/2.8). The morphology and deformation of protocells were characterized by polarized optical microscopy (ZEISS Imager.A2). The surface characteristics of protocells and prototissues were characterized using scanning electron microscopy (SEM, JSM-7900F, JEOL). 2D films were characterized by atomic force microscopy (AFM, Bruker-Fastscan DMFASTSCAN2-SYS). The fluorescence images were obtained using a Leica SP8 laser scanning microscope attached to a Leica DMi 8 inverted fluorescence microscope. The following emission wavelengths ($\lambda_{em}$) were monitored for the corresponding probes: FITC $\lambda_{em}$ = 525 nm, RITC $\lambda_{em}$ = 566 nm, Pyrene $\lambda_{em}$ = 485 nm. Image analysis was performed with ImageJ software. The fluorescence recovery after photobleaching (FRAP) assays were conducted using a FRAP module. One pre-bleach frame (1024 × 1024 pixels) of the labeled CNF/PDDA microcapsule was first recorded at 5% laser power with a 525-nm laser. Afterward, bleaching was performed for 400 s at 100% power of the 525-nm laser by defining the specified region in the FRAP module. The fluorescence recovery was recorded for 300 post-bleach frames (each

frame with an interval of 4 s) at 5% laser power with the 525-nm laser. The intensity of fluorescence was determined by LASX software (Leica). Fourier transform infrared (FTIR) spectra of GO and GO-Pyrene were recorded on a Nicolet 6700 FTIR spectrometer. Raman spectra were acquired on a Renishaw inVia Reflex Raman microscope using a 514 nm excitation laser. Hydrodynamic diameters of PDDA samples were measured by dynamic light scattering (DLS) on a Malvern Zetasizer Nano ZS90 at 25 °C. X-ray photoelectron spectroscopy (XPS) measurements were performed on a Thermo Scientific ESCALAB 250 spectrometer using Al Kα radiation.

### Preparation of protocells and prototissues

For the preparation of protocells, typically, an aqueous phase of 0.6 wt % CNF (or 0.6 wt% CNF + 0.2 wt% GO) was pumped through the inner needle of a coaxial needle at a flow rate of 30 mL h$^{-1}$ using a syringe pump. Compressed nitrogen gas, regulated by a rotameter to a flow rate of 1.0 L min$^{-1}$, was connected to the outer needle of the coaxial setup. The shear force exerted by the outer gas flow fragmented the extruded phase into uniform microdroplets. These microdroplets subsequently fell into the aqueous collecting bath containing 0.5 wt% PDDA and 0.1 wt% Triton X-100 to form CNF/PDDA microcapsules (or CNF-GO/PDDA microcapsules). The produced microcapsules were transferred into water and washed thoroughly, with the water being replaced repeatedly over 10 cycles to ensure complete removal of residual components.

For the preparation of prototissues, an aqueous phase containing 0.6 wt% CNF (or 0.6 wt% CNF + 0.2 wt% GO) and 0.3 M NaCl was extruded into an aqueous collection bath containing 0.5 wt% PDDA and 0.1 wt% Triton X-100 to produce CNF/PDDA microcapsules (or CNF-GO/PDDA microcapsules) (extrusion rate = 30 mL h$^{-1}$, gas flow rate = 1.0 L min$^{-1}$). In the absence of molds, when the needle position was fixed, the microcapsules settled to the bottom of collection bath, where they spontaneously accumulated and adhered to one another to form a macroscopic, self-supporting conical prototissue. Arbitrarily shaped prototissues could also be constructed by manually moving the needle during gas-spraying, but the precision of resulting shapes was relatively limited. When molds were used, a patterned mold was pre-positioned in collection bath, and microcapsules were deposited into the mold by manually moving the needle during gas-spraying process, where they adhered to each other. After depositing the required amount of microcapsules, the mold was lifted to yield prototissues that faithfully replicated the mold geometry. Regardless of mold use, the constructed prototissues were transferred to deionized water and washed by repeated water replacement for 10 cycles to remove residual components.

All patterned molds used in this work were commercially fabricated open-frame molds, consisting of a hollow frame with a defined height and open top/bottom. The molds can be made from polytetrafluoroethylene (PTFE), polypropylene (PP), polystyrene (PS), high-density polyethylene (HDPE), stainless steel, aluminum alloy, or hydrophobically modified glass. Owing to the weak interfacial interactions between these mold materials and prototissues, demolding was readily achieved by simply lifting the mold, allowing the prototissues inside to retain their shape intact.

### Reporting summary

Further information on research design is available in the Nature Portfolio Reporting Summary linked to this article.

## Data availability

Data supporting the findings of this study are available within the paper, its Supplementary Information files and from corresponding authors upon request. Source data for all figures in the main text and Supplementary Information are provided with this paper.

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

## Acknowledgements

This work was supported by the National Natural Science Foundation of China (52322310 and 22372005, both to S.S.; 52221006 to F.-J.X.) and the Beijing Natural Science Foundation (2262070 to S.S.).

## Author contributions

W.F. and S.S. conceived the project. W.F. and S.S. designed the study. W.F. performed the experiments. W.F. analyzed the data. P.L., X.L., Z.W., M.C. and Y.H. assisted with experiments, characterization and discussion of the results. S.S. and F.J.X. supervised the project. W.F. wrote the first draft of the manuscript. F.J.X. and S.S. revised the manuscript. All authors discussed the results, commented on the manuscript and approved the final version of the manuscript.

## Competing interests

The authors declare no competing interests.
