## [Transparent Peer Review file · Nature Communications]

One-Step Construction of Robust Protocells and Prototissues in Water

Corresponding Author: Professor Shaowei Shi

Version 1:

Reviewer comments:

Reviewer #1

(Remarks to the Author)

In this study, the authors present a novel diffusion-inhibited complexation strategy that enables low-cost, high-throughput fabrication of protocells and precise, large-scale 3D construction of prototissues. This simple, fully aqueous method yields protocells and prototissues with excellent mechanical strength. The resulting prototissues reach sizes exceeding 10 cm, significantly larger than the millimeter-scale structures reported previously, addressing a long-standing challenge in the field. The permeability and chemical communication of the protocells, as well as the osmotic-pressure-induced deformation and buoyancy-driven motion of the prototissues were thoroughly investigated. This work provides a promising platform for future applications in areas such as soft robotics, biomedical devices and tissue engineering. Therefore, I recommend the publication of the manuscript in Nature Communications after minor revisions.

1. The system used by the authors appears to be quite different from conventional aqueous two-phase systems (ATPS), such as the widely studied PEG-dextran system. In comparison to ATPS, what are the advantages of achieving compartmentalization in pure water? It would also be helpful if the authors could clarify whether spontaneous phase separation occurs upon direct mixing of CNF and PDDA aqueous solutions.
2. While the effects of gas flow rate and extrusion rate on the size of the microcapsules have been investigated. It is recommended to assess the influence of CNF and PDDA concentrations on microcapsule formation and size.
3. The prepared prototissues exhibit excellent mechanical strength in both water and air, which is very impressive. Would the prototissues still maintain their structural stability after complete removal of water from both the internal and external prototissues?
4. The pH values of both CNF and PDDA solutions should be stated in the manuscript, as the pH is a critical role in the strength of electrostatic interactions between CNF and PDDA. In addition, the influence of pH on microcapsule formation is recommended to be explored.
5. Morphological characterization of CNF such as AFM or TEM should be provided.

Reviewer #2

(Remarks to the Author)

In this work, Weixiao Feng et al. present a microfluidic method to construct protocells and assemble prototissues with complex 3D architecture. While the methodology is novel and interesting, several aspects of its functioning principles remain unclear and require more thorough investigation before publication. Moreover, beyond the novelty of the production method for protocells and prototissues, the work does not present a significant conceptual advancement in the protocell/prototissue engineering field, as it primarily reproduces results that have already been demonstrated by established leaders in the field such as Bayley, Mann, and Han. Given the limited conceptual progress, issues with the methodology's working principles, and inadequate scholarly presentation, I do not recommend publication of this manuscript in Nature Communications.

Below are specific comments that may help improve the rigor and significance of the work:

Materials Characterization and Methodology

1. Essential details regarding CNF and PDDA are missing, which is fundamental for result reproduction. The authors should report the molecular weight and dispersity of all polymers used, their hydrodynamic radius in the working solvent, and degree of branching. Additionally, why is cellulose negatively charged? Since cellulose and PDDA interact electrostatically,

how many charges per polymer chain do they carry?

2. The authors refer to their objects as "microcapsules," but "microspheres" would be more appropriate since they are not hollow structures.
3. The role of Triton-X should be clarified: what concentration was used, and what surface tension is required to achieve microcapsule formation? Can other surfactants be employed, particularly those that are completely biocompatible?
4. The statement "By varying extrusion rate or gas flow rate, the size of microcapsule could be controlled" contradicts Supplementary Figure 3a, which clearly shows that extrusion rate does not influence microcapsule size. This discrepancy should be corrected.
5. The microcapsule structure requires further exploration. Where does PDDA localize? Does it permeate and form a network inside the microsphere, which could explain their resilience and reduced polymer chain mobility?

Permeability Studies

6. The statement "This suggested the poor semi-permeability of membrane, leading to a rapid equilibrium of osmotic pressure across the microcapsules" lacks clarity. It is unclear how membrane permeability can be assessed by placing microcapsules in high-concentration glucose or dextran solutions and observing shrinkage behavior. This requires further explanation.
7. For the permeability experiment using dextran (Supplementary Figure S7), the partition coefficient of FITC-tagged dextran should be provided. Complete characterization of the dextran (molecular weight, dispersity, degree of labeling) is also necessary. Importantly, a molecular weight cut-off for the microspheres should be estimated.
8. There is confusion regarding the statement "It should be noted that if the microcapsules were sufficiently rigid, the shrinkage induced by the hypertonic solution could be suppressed" when earlier the authors mention "no shrinkage of microcapsules was observed within 24 h." This apparent contradiction requires clarification.

Characterization Issues

9. The authors tagged GO with pyrene but did not explain the attachment method or characterize the product. How can they confirm that pyrene is covalently bound rather than physisorbed on GO, which might detach in complex solutions containing diverse polymers and salts? Control experiments are required to validate the conclusions regarding GO localization in microspheres.
10. There is an apparent contradiction regarding GO's effect on permeability. The authors report that GO hinders glucose permeability, with smaller GO amounts leading to faster glucose permeation. However, in earlier experiments (lines 106107), they state that permeability is poor in GO's absence. This needs clarification.

Coacervate Formation and Prototissue Assembly

11. In the discussion from line 213, when using polyethyleneimine, how can the authors confirm coacervate formation rather than simple polymer aggregates formed by electrostatic interactions in a molecularly crowded environment? This claim requires supporting data. Additionally, what is an "egg-shaped coacervate"?
12. The prototissue preparation lacks sufficient detail. The Materials and Methods section only states: "For the preparation of prototissues, 0.3 M electrolyte (e.g., NaCl) was added into the extrusion solution, while other parameters remained unchanged." No explanation of how different shapes are achieved is provided. The term "extrusion droplets" requires definition. Details about the molds (material, fabrication process, filling method, removal procedure) are absent.
13. The nature of microsphere-microsphere adhesion remains unclear. Why does a thicker, more relaxed membrane promote adhesion? While the authors attempt an explanation, this fundamental aspect requires precise understanding supported by experimental data.

Characterization and Applications

14. The authors demonstrate various possibilities that have already been reported in the literature using their new prototissue types but provide insufficient characterization. Detailed mechanical property analysis of protocells and prototissues is necessary to support claims about mechanical properties and bending mechanics of layered prototissues. What forces are exerted during bending?
15. For the buoyant prototissue: what is the intended purpose and biological relevance? What is the buoyancy rate and its dependence on experimental parameters?

Biocompatibility Claims

16. The authors claim biocompatibility in their conclusions without providing supporting data. Triton-X usage could be problematic in this regard. While an in vitro cytotoxicity assay is mentioned in the experimental section, no data are presented in the main text.

Recommendation

The manuscript requires substantial improvements in experimental design, data presentation, and mechanistic understanding before it can meet the publication standards of Nature Communications.

Reviewer #3

(Remarks to the Author)

The authors describe the construction of prototissues via adhesive interaction between microgel compartments fabricated from PDDA and CNF. The membrane permeability of the compartments were studied and modulated by inclusion of GO. They also illustrated how osmotic changes can help regulate the morphology of the microgel compartments irreversibly via cross linking of the interior or reversibly via delayed permeability. They further went on to study a two-enzyme cascade

within the CNF-GO/PDDA microcapsules and show chemomechanical actuation of the prototissues mediated by osmotic changes and bubble mediated buoyancy.

The comments are as follows:

- 1) The primary concern is regarding novelty – as a significant part of the paper mostly deals with preparation of the protocells and their osmotic response along with enzyme cascade studies. These are routine studies in protocell field and the authors do well to thoroughly characterize their new protocell model/chasis but these do not add significant novelty to the work.
- 2) The advantage of a prototissue is that it has compartments which can host a reaction to induce morphological or functional output. In the current design, the authors have used the osmotic response of the protocells and bubble mediated buoyancy to produce actuation. The osmotic pressure mediated actuation due to differential response in the two layer tissue was also illustrated in earlier studies (G. Villar et al. Science 2013, 340, 48-52). The buoyancy-based actuation of prototissues which the authors have shown is quite interesting as it is mediated by reactions within the protocells. However, this reviewer would like to see not just actuation but also recovery and re-actuation over multiple cycles which would then constitute a significant advance in the field.
- 3) One suggestion would be to use Glucose oxidase mediated consumption of the oxygen bubble to execute multiple cycles of actuation or formulate a different strategy for dissolution of the bubble for recovery of the actuated tissue.
- 4) The authors have studied the shrinkage of the CNF-GO/PDDA protocells with PEI and PDDA. It is seen that PDDA 100 kDa permeates and cross links but does not form a yolk structure but PEI forms a yolk structure which constitutes a highly cross linked network. Author could conduct FRAP studies to understand the condensation mediated changes in fluidity of the interior and also clarify the why there is difference in morphological outcomes for PEI and PDDA.
- 5) the facile construction of the prototissue due to sticking interactions at high electrolyte concentrations has the limitation of lacking programmability of the internal structure of the prototissue – in terms of self organization of different protocells to form complex hierarchical structures. This reviewer would like to see the potential of this prototissue design in terms of soft robotics, where these prototissues can manipulate macroscopic objects or these actuatable tissues can be used as artificial muscle to operate a gripper like structure. These experiments would add significant novelty to the manuscript.
- 6) Another general comment would be that it may be investigated what is the advantage of using a prototissue over an actuatable gel for carrying out these tasks.

Version 2:

Reviewer comments:

Reviewer #1

(Remarks to the Author)

The questions have been well addressed and it is recommended for publication.

Reviewer #3

(Remarks to the Author)

The authors have significantly addressed many of the points raised by this reviewer which has increased the scope of the work. One key distinguishing point noted in the revised manuscript is the mechanical strength of the fabricated prototissues which is quite commendable.

Minor points of concern remain which are discussed below:

- 1) The key focus of the manuscript is fabrication of prototissues and in order to mimic prototissues it is essential to have a membrane permeability cut-off which can encapsulate enzymes. The higher permeability of the CNF-GO/PDDA microcapsules fabricated can only entrap small enzymes such as Horse Radish Peroxidase (44 kDa) as also seen in their data where the peroxidase is only found entrapped in the membrane not in the lumen. The permeability of the microcapsules has been adjusted by adding the GO – can this be taken forward more to get more stringent molecular weight cut-off to be able to encapsulate enzymes? If yes, add relevant data or these limitations and potential ways to overcome this may be discussed in the conclusion.
- 2) The authors have fabricated a soft gripper powered by buoyant forces. The limitation of such design is that the forces are only acting along the vertical axis. The osmotic actuation is not enzyme driven and hence not as interesting, but the buoyant design seems limited in its outlook. These limitations may be discussed and future plan for improvements may be discussed in the conclusion.

Manuscript ID: NCOMMS-25-49805A-Z

Title: One-Step Construction of Robust Protocells and Prototissues in Water

Authors: Weixiao Feng, Peifan Li, Xin Li, Ziwei Wang, Min Chen, Yang Hu, Fu-Jian Xu*, Shaowei Shi*

Dear editor:

We would like to thank the reviewers for their detailed feedback to improve our submission. We have considered each comment carefully and revised our manuscript to address the issues raised.

Responses to Reviewer #1

Reviewer #1 (Remarks to the Author):

In this study, the authors present a novel diffusion-inhibited complexation strategy that enables low-cost, high-throughput fabrication of protocells and precise, large-scale 3D construction of prototissues. This simple, fully aqueous method yields protocells and prototissues with excellent mechanical strength. The resulting prototissues reach sizes exceeding 10 cm, significantly larger than the millimeter-scale structures reported previously, addressing a long-standing challenge in the field. The permeability and chemical communication of the protocells, as well as the osmotic-pressure-induced deformation and buoyancy-driven motion of the prototissues were thoroughly investigated. This work provides a promising platform for future applications in areas such as soft robotics, biomedical devices and tissue engineering. Therefore, I recommend the publication of the manuscript in Nature Communications after minor revisions.

1. The system used by the authors appears to be quite different from conventional aqueous two-phase systems (ATPS), such as the widely studied PEG-dextran system. In comparison to ATPS, what are the advantages of achieving compartmentalization in pure water? It would also be helpful if the authors could clarify whether spontaneous phase separation occurs upon direct mixing of CNF and PDDA aqueous solutions.

Response: We appreciate the comment from the reviewer. Conventional ATPS is a mixture of two polymers, or a polymer and a salt, which phase-separates to form two immiscible aqueous phases. To achieve stable compartmentalized microcapsules, it is usually necessary to introduce additional solid particles or oppositely charged polyelectrolytes to stabilize the interface (*Langmuir* **2012**, *28*, 5921-5926; *ACS Macro Lett.* **2016**, *5*, 666-670). In contrast, our system enables the one-step fabrication of microcapsules based on two aqueous solutions using the DIC method. This strategy

greatly simplifies the preparation process, requires fewer components, and reduces overall cost.

We compared the mixing behaviors of the dextran-PEG ATPS and the CNF-PDDA DIC system. As shown in Fig. R1, in the ATPS, mixing 20 wt% dextran solution with 15 wt% PEG solution results in spontaneous separation into immiscible dextran-rich and PEG-rich phases. After vigorous shaking to homogenize the mixture, it re-separates into two phases to minimize interfacial energy. In the DIC system, when 0.5 wt% PDDA solution is brought into contact with 0.8 wt% CNF solution, electrostatic complexation occurs instantaneously, forming a thin film that suppresses further diffusion and mixing. After vigorous shaking, the system undergoes polyelectrolyte complex flocculation rather than forming the two aqueous phases observed in the ATPS, indicating that compartmentalization in our system originates from interfacial film formation rather than from bulk phase separation.

Fig. R1. Mixing behaviors of (a) dextran-PEG ATPS and (b) CNF-PDDA DIC system. Scale bar, 0.5 cm.

2. While the effects of gas flow rate and extrusion rate on the size of the microcapsules have been investigated. It is recommended to assess the influence of CNF and PDDA concentrations on microcapsule formation and size.

Response: We appreciate the comment from the reviewer. We systematically investigated the influence of CNF and PDDA concentrations on microcapsule formation at a fixed extrusion rate of 30 mL h^{-1} and a compressed gas flow rate of 1.0 L min^{-1} . As shown in Fig. R2, when the CNF concentration is fixed, increasing the PDDA concentration raises the viscosity of the collection bath, which prevents CNF-containing droplets from entering the bath smoothly and thus hinders

microcapsule formation. On the other hand, at a fixed PDDA concentration, increasing the CNF concentration enhances the mechanical strength of microcapsules, thereby facilitating their formation. Within the concentration range that yields microcapsules, the size remains nearly constant at approximately 800 μm , indicating that concentration does not significantly influence microcapsule dimensions.

We have added Fig. R2 in the revised Supplementary Information as Supplementary Fig. 6, along with a brief discussion.

Fig. R2. Influence of CNF and PDDA concentrations on microcapsule formation. [Triton X-100] = 0.1 wt%; extrusion rate = 30 mL h⁻¹; gas flow rate = 1.0 L min⁻¹.

3. The prepared prototissues exhibit excellent mechanical strength in both water and air, which is very impressive. Would the prototissues still maintain their structural stability after complete removal of water from both the internal and external prototissues?

Response: We appreciate the reviewer's comment. As shown in Fig. R3, freeze-drying the CNF–GO/PDDA prototissues yields a highly dehydrated, tissue-like material that retains its structural integrity. Even under a 200 g load, the structure remains intact, demonstrating the excellent mechanical stability of the prototissue.

We have added a brief discussion in the revised manuscript as follows: “Freeze-drying the CNF–GO/PDDA prototissue yielded a fully dehydrated, tissue-like material that retains its structural integrity and was capable of supporting external loads”. Fig. R3 has been included in the revised

Supplementary Information as Supplementary Fig. 27.

Fig. R3. Digital images showing the excellent mechanical stability of freeze-dried CNF–GO/PDDA prototissue. Scale bar, 1 cm.

4. The pH values of both CNF and PDDA solutions should be stated in the manuscript, as the pH is a critical role in the strength of electrostatic interactions between CNF and PDDA. In addition, the influence of pH on microcapsule formation is recommended to be explored.

Response: We appreciate the comment from the reviewer. We have stated the pH values in the revised Supplementary Information, where the 0.6 wt% CNF and 0.5 wt% PDDA solutions exhibit pH values of approximately 7.0 and 5.0, respectively. In fact, we did not intentionally adjust the pH of either solution during the experiments. We fully agree with the reviewer that pH plays a critical role in governing the strength of electrostatic interactions between CNF and PDDA, and we have systematically investigated the influence of pH (1.0-13) on microcapsule formation. As shown in Fig. R4, microcapsules can be successfully produced over a wide pH range. At extremely low pH (pH = 1.0), however, extensive protonation of the carboxyl groups on CNF weakens electrostatic interactions with PDDA, resulting in failure of microcapsule formation. Within the pH range that yields microcapsules, the capsule size remains nearly constant at approximately 800 μm , indicating that pH does not significantly influence microcapsule dimensions.

We have added Fig. R4 in the revised Supplementary Information as Supplementary Fig. 7, along with a brief discussion.

Fig. R4. Influence of pH on microcapsule formation. [CNF] = 0.6 wt%; [PDDA] = 0.5 wt%; extrusion rate = 30 mL h⁻¹; gas flow rate = 1.0 L min⁻¹.

5. Morphological characterization of CNF such as AFM or TEM should be provided.

Response: We appreciate the comment from the reviewer, and have provided TEM characterization of CNF. As shown in Fig. R5, CNF is a fibrous, one-dimensional nanomaterial with diameters of 10-20 nm and lengths of several micrometers. Fig. R5 has been included in the revised Supplementary Information as Supplementary Fig. 1.

Fig. R5. TEM images of CNF.

Responses to Reviewer #2

Reviewer #2 (Remarks to the Author):

In this work, Weixiao Feng et al. present a microfluidic method to construct protocells and assemble prototissues with complex 3D architecture. While the methodology is novel and interesting, several aspects of its functioning principles remain unclear and require more thorough investigation before publication. Moreover, beyond the novelty of the production method for protocells and prototissues, the work does not present a significant conceptual advancement in the protocell/prototissue engineering field, as it primarily reproduces results that have already been demonstrated by established leaders in the field such as Bayley, Mann, and Han. Given the limited conceptual progress, issues with the methodology's working principles, and inadequate scholarly presentation, I do not recommend publication of this manuscript in Nature Communications. The manuscript requires substantial improvements in experimental design, data presentation, and mechanistic understanding before it can meet the publication standards of Nature Communications.

Response: We appreciate the reviewer's comment and recognition of our method for preparing protocells and prototissues. As discussed in the introduction, beyond the novelty of the production method, this work also addresses several key challenges in this field, including the large-scale preparation of protocells/prototissues in aqueous media, achieving excellent mechanical strength and stability, and enabling precise shaping of prototissues. We fully acknowledge the pioneering contributions of Bayley, Mann and Han, but we believe our study makes substantive progress on several long-standing issues. In the revised manuscript, we have added substantial experimental data and discussion, addressing all reviewer questions and including additional results such as the periodic motion of prototissues, the preparation of gripper-like prototissues, and their ability to manipulate macroscopic objects, which further strengthen the significance of our work. We hope that the revised manuscript meets the reviewer's high expectations.

To further clarify the novelty of our contribution, we summarize below several bottlenecks and challenges identified in review articles by multiple groups, including Prof. Han's.

Prof. Gobbo et al. (*Biochem. Soc. Trans.* **2020**, *48*, 2579–2589; Prof. Gobbo was a postdoctoral fellow in the research group of Professor Stephen Mann):

- “To date, prototissues have been assembled in the millimetre and submillimetre size range using a variety of techniques...A materials chemistry approach would allow the advancement

of current prototissue construction technologies and enable the precise assembly of protocells into large-scale materials (>1 cm in size) that are robust, free-standing, and stable in water media. This is critical for future technological applications of prototissues and would represent a major achievement in the field.”

- “To date, research has been focusing on the development of methodologies to assemble protocells together in a controlled and reproducible manner. However, if we want to develop prototissue-based technologies, it is of paramount importance that we start advancing the prototissue’s mechanical properties. In this regard, current studies have been limited to the characterisation of the prototissue’s tensile strength and elastic modulus..It is now critical to construct protocellular materials with a range of mechanical properties that mimic those of living tissues and perhaps even outperform them.”

Prof. Han et al. (*Ac/v. Funct. Mater.* **2024**, *34*, 2405823):

- “Although prototissues with precise spatial structures can be assembled with micrometer and millimeter ranges, they are still small compared to tissues for tissue transplantation. To assemble large-scale prototissues, the long-distance signaling, synergistic effects between artificial cells, and collective functions will face enormous challenges.”

Prof. Lin et al. (*Ac/v. Healthcare Mater.* **2025**, *14*, 2500376):

- “Despite significant progress in the development of prototissues, several challenges remain... 2) Natural tissues exhibit a broad range of mechanical strength due to the ECM composition (e.g., collagen, elastin) and cellular interactions. The mechanical stability of prototissues or protocell assemblies is typically low; as such, the incorporation of synthetic hydrogels, scaffolds, or artificial ECM needs to be considered for structural support ...5) Engineering protocell assemblies into large-scale, functional 3D structures with batch-to-batch reproducibility remains challenging. Current fabrication techniques, including microfluidics and 3D printing, struggle to precisely control spatial organization at a tissue-like scale.”

Below are specific comments that may help improve the rigor and significance of the work:

Materials Characterization and Methodology

1. Essential details regarding CNF and PDDA are missing, which is fundamental for result reproduction. The authors should report the molecular weight and dispersity of all polymers used,

their hydrodynamic radius in the working solvent, and degree of branching. Additionally, why is cellulose negatively charged? Since cellulose and PDDA interact electrostatically, how many charges per polymer chain do they carry?

Response: We appreciate the comment from the reviewer. CNF is not traditional linear polymer but cellulose-based polysaccharide nanomaterial that contain both crystalline and amorphous regions. Therefore, parameters such as molecular weight, dispersity, hydrodynamic radius, and degree of branching are not appropriate descriptors for CNF. Here we characterized CNF using TEM. As shown in Fig. R6a,b, CNF are fibrous, one-dimensional nanomaterials with diameters of 10-20 nm and lengths of several micrometers. The negative charge of CNFs arises from the 2,2,6,6-tetramethylpiperidiny-1-oxyl (TEMPO) oxidation process used in their preparation, during which a fraction of the primary C6 hydroxyl groups on the cellulose backbone is converted into carboxyl groups (–COOH) (*Biomacromolecules* **2007**, *8*, 2485-2491; *Nanoscale* **2011**, *3*, 71-85). According to the supplier's specifications, the carboxylate content of the CNFs is 1.2-3.0 mmol COOH per gram of CNF.

PDDA, on the other hand, is a linear polymer without branching (Fig. R6c). In this work, three PDDA samples with different molecular weights (M_w : 100-200 kDa, 200-350 kDa, 400-500 kDa) were used, all purchased from Shanghai Aladdin Biochemical Technology Co., Ltd. Detailed information on molecular weights is provided at the end of the reply letter. Because commercial PDDA typically has a broad molecular-weight distribution, the dispersity is of limited significance and generally not reported by suppliers. The hydrodynamic sizes of PDDA in deionized water (25 °C) were measured by dynamic light scattering (DLS) at 0.01 mg mL⁻¹, yielding values of 23.4, 29.4, and 40.5 nm for the three samples (Fig. R7). Moreover, since each repeat unit carries one charged quaternary ammonium group, the number of charges per polymer chain corresponds to the degree of polymerization, where the weight-average degrees of polymerization for the three PDDA samples are ~600-1200, ~1200-2200, and ~2500-3100, respectively.

We have stated the molecular weights and hydrodynamic sizes of PDDA in *Materials* section. Fig. R6 has been included in the revised Supplementary Information as Supplementary Fig. 1.

Fig. R6 **a,b** TEM images of CNF. **c** Chemical structure of PDDA.

Fig. R7 Characterization of PDDA hydrodynamic sizes in water using DLS.

2. The authors refer to their objects as "microcapsules," but "microspheres" would be more appropriate since they are not hollow structures.

Response: We appreciate the reviewer's comment. We prefer the term "microcapsules" rather than "microspheres" because the internal and surface structures of the CNF (or CNF-GO)/PDDA assemblies differ significantly. Taking CNF/PDDA microcapsules as an example, CLSM and SEM images show that CNF is mainly distributed at the surface and electrostatically complexes with PDDA to form a dense membrane, while the interior consists of a loose CNF network (Fig. R8a-d). Although this structure is not completely hollow, its shrinkage behavior under osmotic pressure closely resembles that of conventional microcapsules.

As a comparison, we fabricated sodium alginate–Ca²⁺ hydrogel microspheres using a gas–liquid microfluidic approach by extruding sodium alginate droplets into a CaCl₂-containing aqueous bath (*Adv. Sci.* **2019**, *9*, 1802342; *Adv. Mater. Technol.* **2023**, *5*, 2201559). As shown in Fig. R8e-h, when alginate is labeled with rhodamine-B-isothiocyanate (RITC), CLSM shows uniform fluorescence throughout both the interior and the surface, indicating a homogeneous structure. SEM images of

freeze-dried microspheres further confirm structural uniformity from surface to interior. These results confirm that the assemblies formed from CNF (or CNF-GO) and PDDA are more accurately classified as microcapsules rather than microspheres.

Fig. R8. **a**, CLSM images of a CNF/PDDA microcapsule containing CNF-FITC and corresponding fluorescence intensity profile (**b**, dashed line in **a**). Scale bar, 200 μm . **c,d**, SEM images of a ruptured, freeze-dried CNF/PDDA microcapsule, showing the membrane (**c**, scale bar, 100 μm) and the internal fibrillar network (**d**, scale bar, 10 μm). **e**, CLSM image of Sodium alginate- Ca^{2+} hydrogel microspheres containing Alg-RITC and corresponding fluorescence intensity profile (**f**, dashed line in **e**). Scale bar, 200 μm . **g,h**, Freeze-dried SEM images reveal a continuous, uniform gel network from surface to interior. Scale bar in **g**, 100 μm , scale bar in **h**, 10 μm .

3. The role of Triton-X should be clarified: what concentration was used, and what surface tension is required to achieve microcapsule formation? Can other surfactants be employed, particularly those that are completely biocompatible?

Response: We appreciate the comment from the reviewer. Triton X-100 serves to reduce the air-water surface tension of the collection bath, enabling the droplets to pass through the interface and undergo DIC to form microcapsules. Without Triton X-100, or when its concentration is very low, intact CNF/PDDA microcapsules cannot be produced. The concentration of Triton X-100 used in our work is 0.1 wt%. To clarify the relationship between surface tension and microcapsule formation, we measured the surface tension of Triton X-100 solutions at different concentrations and prepared microcapsules accordingly. As shown in Fig. R9, microcapsules can be successfully formed when the surface tension is below $\sim 45 \text{ mN m}^{-1}$ (corresponding to a Triton X-100 concentration of $\sim 6 \times$

10^{-4} wt%). When the surface tension exceeds 45 mN m^{-1} , CNF-containing droplets are unable to pass through the interface and therefore fail to transform into microcapsules.

In addition to Triton X-100, we also tested four widely used, nonionic and biocompatible surfactants, including Pluronic F68, Pluronic F127, Tween-20, and Solutol HS-15. All of them enabled successful microcapsule formation at a concentration of 0.1 wt%, indicating that the type of surfactant does not affect the ability to produce microcapsules (Fig. R10).

We have made a brief discussion in the revised manuscript as follows: “Triton X-100 was used to reduce the air-water surface tension of the collection bath, facilitating droplets passing through the surface and transforming into microcapsules by DIC. Without Triton X-100, or when its concentration was very low, the high interfacial tension prevented the formation of intact CNF/PDDA microcapsules. In addition to Triton X-100, surfactants including Pluronic F68, Pluronic F127, Tween-20, and Solutol HS-15 also enabled successful microcapsule formation”. Fig. R9 and Fig. R10 have been included in the revised Supplementary Information as Supplementary Fig. 3 and 5.

Fig. R9. Equilibrium air-water surface tension of the collection bath as a function of Triton X-100 concentration and the resulting effect on microcapsule formation.

Fig. R10. a-d Digital images showing the successful fabrication of CNF/PDDA microcapsules using four different surfactants including Pluronic F68, Pluronic F127, Tween-20, and Solutol HS-15. [CNF] = 0.6 wt%; [PDDA] = 0.5 wt%; [Surfactant] = 0.1 wt%; extrusion rate = 30 mL h⁻¹; gas flow rate = 1.0 L min⁻¹. Scale bar, 1 cm.

4. The statement "By varying extrusion rate or gas flow rate, the size of microcapsule could be controlled" contradicts Supplementary Figure 3a, which clearly shows that extrusion rate does not influence microcapsule size. This discrepancy should be corrected.

Response: We appreciate the comment from the reviewer. We have removed the inaccurate description from the manuscript and have made a brief discussion in the revised Supplementary Information as follows: "At a fixed gas flow rate of 1.0 L min⁻¹, varying the extrusion rate from 5.0 to 50 mL h⁻¹ produces microcapsules with nearly constant sizes of ~ 800 μm. In contrast, at a fixed extrusion rate of 30 mL h⁻¹, increasing the gas flow rate from 0.8 to 1.6 L min⁻¹ reduces the microcapsule size from ~1000 to ~ 400 μm. These results indicate that microcapsule size is governed primarily by the gas flow rate rather than the extrusion rate."

5. The microcapsule structure requires further exploration. Where does PDDA localize? Does it permeate and form a network inside the microsphere, which could explain their resilience and reduced polymer chain mobility?

Response: We appreciate the comment from the reviewer. PDDA is primarily localized at the membrane of the microcapsules rather than within their interiors. To directly visualize the distribution of PDDA, we labeled PDDA with rhodamine-B-isothiocyanate (RITC) and used it to produce CNF/PDDA microcapsules. As shown in Fig. R11, the CLSM image reveals that PDDA is predominantly concentrated on the microcapsule surface and does not permeate or form a network inside the capsule. Therefore, the resilience of the microcapsules and the reduced polymer chain mobility arise mainly from the internal CNF network and the composite film formed between CNF (or CNF-GO) and PDDA.

We have added a brief discussion in the revised manuscript as follows: “Using rhodamine-B-isothiocyanate (RITC)-labelled PDDA (PDDA-RITC) as a fluorescent probe, it was observed that PDDA was fully concentrated on the membrane and did not permeate into the interior.” The synthesis of PDDA-RITC has been incorporated into the *Supplementary Methods* section (*Synthesis of fluorescently labelled dextran, CNF, enzymes and PDDA*). Fig. R11 has been included in the revised Supplementary Information as Supplementary Fig. 11.

Fig. R11. CLSM images of a CNF/PDDA microcapsule containing PDDA-RITC and corresponding fluorescence intensity profile along the dashed line. Scale bar, 200 μm.

Permeability Studies

6. The statement "This suggested the poor semi-permeability of membrane, leading to a rapid equilibrium of osmotic pressure across the microcapsules" lacks clarity. It is unclear how membrane permeability can be assessed by placing microcapsules in high-concentration glucose or dextran solutions and observing shrinkage behavior. This requires further explanation.

Response: We appreciate the comment from the reviewer. The reviewer is correct that membrane

permeability is not affected by the surrounding high-concentration solution. Our intention was to clarify that the membrane of CNF/PDDA microcapsules has relatively large pores and does not selectively block small molecules or polymers, e.g., glucose or dextran. In other words, the CNF/PDDA membrane does not behave as a semipermeable membrane. As a result, when the microcapsules are placed in a high-concentration glucose or dextran solution, although an osmotic pressure difference initially exists, both water molecules and solutes (glucose or dextran) can freely diffuse across the membrane. This rapid diffusion leads to fast equilibration of osmotic pressure inside and outside the microcapsule, and no shrinkage behavior is observed. In contrast, if the membrane were semipermeable and able to block glucose or dextran, only water would diffuse outward while the solute remained outside, resulting in microcapsule shrinkage.

We have revised the manuscript as follows: “This indicates that the membrane of the CNF/PDDA microcapsule does not selectively block glucose or dextran, and instead both solutes freely diffuse into the microcapsule, leading to rapid equilibrium of osmotic pressure across the membrane.

7. For the permeability experiment using dextran (Supplementary Figure S7), the partition coefficient of FITC-tagged dextran should be provided. Complete characterization of the dextran (molecular weight, dispersity, degree of labeling) is also necessary. Importantly, a molecular weight cut-off for the microspheres should be estimated.

Response: We appreciate the comment from the reviewer. Because the concentration of FITC-labeled dextran (Dex-FITC) inside and outside the CNF/PDDA microcapsules is proportional to the fluorescence intensity, the partition coefficient (PC) can be determined from the ratio of internal to external fluorescence intensity. As shown in Fig. R12, using $PC = C_{in}/C_{out} \approx I_{in}/I_{out}$, the PC values of Dex-FITC with molecular weights of 10, 40, 70, 100, and 500 kDa are 0.37, 0.34, 0.34, 0.34, and 0.22, respectively. It should be noted that, although the CNF/PDDA microcapsule membrane does not block Dex-FITC, the interior is filled with a loose CNF network that reduces the accessible free volume for dextran. In addition, negatively charged CNF introduces electrostatic repulsion against the anionic FITC label. Both steric exclusion and electrostatic effects lead to a lower Dex-FITC content inside the microcapsules compared to the exterior.

We have added Fig. R12 to the revised Supplementary Information as Supplementary Fig. 13, along with a brief discussion.

Fig. R12. a-e, Confocal fluorescence images of CNF/PDDA microcapsules after equilibration for 4 h in dextran solutions containing trace amounts of FITC-dextran. Scale bars, 200 μm . f, Summary of partition coefficients (PCs) of dextran, determined using FITC-dextran as a fluorescent tracer, where the PC is defined as the ratio of internal to external fluorescence intensity. Error bars represent the standard deviation ($n = 3$).

Five dextran samples with different molecular weights (M_w : 10, 40, 70, 100, and 500 kDa) were purchased from J&K Scientific. Detailed information on molecular weights is provided at the end of reply letter. Commercial dextran is a natural polysaccharide produced via microbial fermentation and inherently exhibits a broad molecular-weight distribution. Therefore, suppliers typically provide only the average molecular weight rather than dispersity values. Dex-FITC was prepared by covalently grafting FITC onto commercially purchased dextran. Based on ^1H NMR spectra (Fig. R13), the FITC grafting ratios for the five dextran samples were determined to be 0.0526, 0.0469, 0.0358, 0.0281, and 0.0226 mol FITC per mol glucose (mol%), respectively. As Dex-FITC serves solely as a fluorescent tracer in this study, the grafting density of FITC is not a critical parameter for the main conclusions. Therefore, we provide the detailed grafting ratios in the reply letter rather than incorporating them into the revised manuscript.

For the molecular-weight cut-off of CNF/PDDA microcapsules, the fact that even the largest dextran sample (M_w : 500 kDa) can permeate the membrane indicates that CNF/PDDA microcapsules are not suitable as protocell models. Consequently, estimating a molecular-weight cut-off is not particularly meaningful, as the membrane does not impose effective size exclusion. This is also why we further incorporated graphene oxide into the membrane to enhance molecular selectivity.

Fig. R13. a, Chemical structure of FITC-dextran. b–f, ¹H NMR spectra of FITC-dextran (400 MHz, D₂O, 25 °C).

8. There is confusion regarding the statement "It should be noted that if the microcapsules were sufficiently rigid, the shrinkage induced by the hypertonic solution could be suppressed" when earlier the authors mention "no shrinkage of microcapsules was observed within 24 h." This apparent contradiction requires clarification.

Response: We appreciate the comment from the reviewer. As discussed in the manuscript,

CNF/PDDA microcapsules do not shrink when placed in a neutral glucose or dextran hypertonic solution, but they do shrink in a negatively charged poly(acrylic acid) hypertonic solution. Our intention was to use this shrinkage behavior to illustrate the elastic nature of the membrane. However, as the reviewer pointed out, the original wording could lead to confusion. Therefore, in the revised manuscript, we have removed the sentence “It should be noted that if the microcapsules were sufficiently rigid, the shrinkage induced by the hypertonic solution could be suppressed” and instead discuss the behaviors in glucose/dextran and poly(acrylic acid) solutions separately.

Characterization Issues

9. The authors tagged GO with pyrene but did not explain the attachment method or characterize the product. How can they confirm that pyrene is covalently bound rather than physisorbed on GO, which might detach in complex solutions containing diverse polymers and salts? Control experiments are required to validate the conclusions regarding GO localization in microspheres.

Response: We appreciate the comment from the reviewer. The synthesis of pyrene-labelled GO (GO-pyrene) has been incorporated into the *Methods* section (*Synthesis of fluorescently labelled nanomaterials and polymers*) as follows:

“Synthesis of GO-Pyrene

Graphene oxide (GO, 100 mg) was refluxed in thionyl chloride (50 mL) for 12 h, and excess thionyl chloride was removed by rotary evaporation to afford GO-COCl (91 mg). GO-COCl was then reacted with cysteine (50 mg) in DMF (50 mL) in the presence of triethylamine (0.5 mL) at room temperature for 24 h to yield GO-Cys (80 mg) after filtration, washing (DMF, deionized water, and ethanol), and vacuum drying. GO-Cys was dispersed in DMF/H₂O (50 mL, v/v ≈ 4:1), treated with TCEP (50 mg) for 12 h, and then reacted with pyrene-maleimide (50 mg) at room temperature for 24 h. The product was collected and washed with DMF, deionized water, and ethanol, followed by repeated ultrasonication in fresh ethanol to remove physically adsorbed pyrene-maleimide, and finally vacuum-dried to afford GO-Pyrene (52 mg).”

For the characterization of GO-pyrene, FTIR and Raman spectroscopy were employed to demonstrate that pyrene is covalently bound to GO:

FTIR Analysis. Pristine GO shows the expected bands of oxygenated graphene oxide, including a

broad O–H stretching band ($\sim 3440\text{ cm}^{-1}$), a C=O stretching band of –COOH groups ($\sim 1730\text{ cm}^{-1}$), and C–O/C–O–C vibrations ($\sim 1385\text{ cm}^{-1}$). After functionalization, the spectrum of GO–pyrene displays new features in the amide I/II region (~ 1648 and $\sim 1546\text{ cm}^{-1}$), consistent with the formation of amide linkages. The carbonyl region of GO–pyrene therefore reflects contributions from both carboxyl and amide C=O groups, giving an altered band shape compared to pristine GO. These changes are consistent with the introduction of cysteine–pyrene moieties covalently attached to the GO surface.

Fig. R14. FTIR spectra of pristine GO (red) and GO–pyrene (grey).

Raman Analysis. In the Raman spectra, both GO and GO–pyrene exhibit the characteristic D ($\sim 1350\text{ cm}^{-1}$) and G ($\sim 1590\text{ cm}^{-1}$) bands of graphene oxide. Upon conversion to GO–pyrene, the I_D/I_G ratio increases from 0.85 (GO) to 1.12 (GO–pyrene), which is consistent with an increased defect density/disorder in the sp^2 carbon framework after chemical modification.

Fig. R15. Raman spectra of pristine GO (grey) and GO–pyrene (red) (excitation: 514 nm).

XPS Analysis. The XPS results are consistent with successful formation of GO-Pyrene. In the survey spectrum of GO-Pyrene, in addition to the dominant C 1s and O 1s peaks characteristic of GO, clear N 1s (~400 eV) and S 2p (~163–169 eV) signals are observed. The high-resolution N 1s spectrum can be deconvoluted into components centred at ~399.6 and ~400.2 eV, which are characteristic of amide/amine-type nitrogen introduced by the pyrene linker. The S 2p region shows a low-binding-energy doublet at ~164.1 and ~165.5 eV, assignable to C–S / C–S–C species, together with a higher-binding-energy component near ~168.4 eV that indicates a small fraction of oxidized sulfur, likely arising from partial oxidation of the sulfur-containing linker. The appearance of these N and S signals in the expected binding-energy ranges, absent in pristine GO, strongly supports that pyrene groups have been covalently grafted onto the GO sheets.

Fig. R16. XPS analysis of GO–Pyrene. a, Survey spectrum of GO–Pyrene. b, High-resolution N 1s spectrum. c, High-resolution S 2p spectrum.

To further clarify the localization of GO in microspheres, we dispersed CNF–GO/PDDA microcapsules in dextran and NaCl solutions, using pyrene-labelled GO for fluorescence visualization. As shown in Fig. R17, CLSM images indicate that GO remains predominantly localized at the membrane in both solutions, as evidenced by the weak fluorescence signal inside the microcapsules. This behavior is consistent with that observed in pure water, demonstrating that the external solution environment does not affect the localization of GO.

Fig. R14, Fig. R15 and Fig. R16 have been included in the revised Supplementary Information as Supplementary Fig. 18, 19 and 20. Since incorporating control experiments in Fig. R17 into the main text or Supporting Information would substantially disrupt the overall structure and narrative flow of the manuscript. Therefore, we provide these results and analysis in the reply letter for the reviewer’s evaluation.

Fig. R17. Confocal fluorescence images (left) and corresponding intensity profiles along the dashed lines (right) for CNF-GO/PDDA microcapsules containing GO-pyrene. a,b As-prepared microcapsules. c,d Microcapsules after incubation for 48 h in 1 M NaCl solution. e,f Microcapsules after incubation for 48 h in 1 wt% of dextran-100k. Scale bars, 200 μm .

10. There is an apparent contradiction regarding GO's effect on permeability. The authors report that GO hinders glucose permeability, with smaller GO amounts leading to faster glucose permeation. However, in earlier experiments (lines 106-107), they state that permeability is poor in GO's absence. This needs clarification.

Response: We appreciate the reviewer's comment and apologize for the misunderstanding caused

by our wording (lines 106–107). In fact, the membrane of CNF/PDDA microcapsules has relatively large pores and does not selectively block glucose or dextran. As clarified in our response to Comment 6, we have revised the manuscript as follows: “This indicates that the membrane of CNF/PDDA microcapsules does not selectively block glucose or dextran, and instead both solutes freely diffuse into the microcapsule, leading to rapid equilibrium of osmotic pressure across the membrane.”

Coacervate Formation and Prototissue Assembly

11. In the discussion from line 213, when using polyethyleneimine, how can the authors confirm coacervate formation rather than simple polymer aggregates formed by electrostatic interactions in a molecularly crowded environment? This claim requires supporting data. Additionally, what is an "egg-shaped coacervate"?

Response: We thank the reviewer for this important comment. The reviewer is correct that when positively charged polyethyleneimine (PEI) diffuses into the CNF-GO/PDDA microcapsules, its electrostatic interactions with negatively charged CNF/GO lead to the formation of solid-like aggregates, rather than liquid-like coacervates.

To verify this conclusion, we synthesized FITC-labelled PEI and characterized the spherical structures formed inside the microcapsules using fluorescence recovery after photobleaching (FRAP). As shown in Fig. R18, the fluorescence intensity displayed almost no recovery after photobleaching, indicating that PEI exhibits very limited molecular mobility, which is a hallmark of solid-like aggregates. In contrast, if liquid-like coacervates had formed, the internal fluidity would have resulted in a rapid fluorescence recovery.

We have removed the incorrect and potentially misleading wording in the revised manuscript, such as “egg-shaped coacervate”. In addition, we have added a new discussion on the formation and characterization of the aggregates, as follows: “PEI-0.6k was able to diffuse into the microcapsules and electrostatically interact with CNF/GO. However, in contrast to PDDA-100k, which bears permanently charged quaternary ammonium groups, the protonatable amine groups along the PEI-0.6k chains provided weaker electrostatic interactions. In addition, the much lower molecular weight and shorter chain length of PEI-0.6k limited the spatial extent of electrostatic crosslinking with CNF/GO, resulting in a more localized and mechanically weaker network. Consequently, the PEI-

0.6k-induced crosslinked structure was unable to resist osmotic swelling upon redispersion of the microcapsules in pure water, leading to network disruption, fragmentation, and the formation of small fragments that gradually associated into larger aggregates. Using FITC-labelled PEI-0.6k as a fluorescent probe and characterizing the aggregates by FRAP, we observed almost no fluorescence recovery after photobleaching, indicating the limited mobility of PEI." Fig. R18 has been included in the revised Supplementary Information as Supplementary Fig. 24.

Fig. R18. a, Confocal fluorescence images of a CNF-GO/PDDA microcapsule containing PEI-FITC before bleaching (-1 s), immediately after bleaching (0 s) and after 1200 s recovery, with the bleached region indicated by the arrow. Scale bar, 200 μm. b, The corresponding fluorescence intensity in the bleached region.

12. The prototissue preparation lacks sufficient detail. The Materials and Methods section only states: "For the preparation of prototissues, 0.3 M electrolyte (e.g., NaCl) was added into the extrusion solution, while other parameters remained unchanged." No explanation of how different shapes are achieved is provided. The term "extrusion droplets" requires definition. Details about the molds (material, fabrication process, filling method, removal procedure) are absent.

Response: We appreciate the comment from the reviewer. We have added a detailed description of the prototissue preparation process including the details about the molds in the *Methods* section as follows:

"For the preparation of prototissues, an aqueous phase containing 0.6 wt% CNF (or 0.6 wt% CNF + 0.2 wt% GO) and 0.3 M NaCl was extruded into an aqueous collection bath containing 0.5 wt% PDDA and 0.1 wt% Triton X-100 to produce CNF/PDDA microcapsules (or GO-CNF/PDDA microcapsules) (extrusion rate = 30 mL h⁻¹, gas flow rate = 1.0 L min⁻¹). In the absence of molds, when the needle position was fixed, the microcapsules settled to the bottom of collection bath, where they spontaneously accumulated and adhered to one another to form a macroscopic, self-supporting

conical prototissue. Arbitrary shaped prototissues could also be constructed by manually moving the needle during gas-spraying, but the precision of resulting shapes was relatively limited. When molds were used, a patterned mold was pre-positioned in collection bath, and microcapsules were deposited into the mold by manually moving the needle during gas-spraying process, where they adhered to each other. After depositing the required amount of microcapsules, the mold was lifted to yield prototissues that faithfully replicated the mold geometry. Regardless of mold use, the constructed prototissues were transferred to deionized water and washed by repeated water replacement for 10 cycles to remove residual components.

All patterned molds used in this work were commercially fabricated open-frame molds, consisting of a hollow frame with a defined height and open top and bottom. The molds can be made from polytetrafluoroethylene (PTFE), polypropylene (PP), polystyrene (PS), high-density polyethylene (HDPE), stainless steel, aluminum alloy, or hydrophobically modified glass. Owing to the weak interfacial interactions between these mold materials and prototissues, demolding was readily achieved by simply lifting the mold, allowing the prototissues inside to retain their shape intact.”

The term “extrusion droplets” was intended to describe droplets formed when the aqueous phase is extruded and subsequently gas-sheared. However, to avoid confusion, in the revised manuscript we have replaced “extrusion droplets” with “extruded aqueous phase,” which more clearly describes the process without introducing unnecessary terminology.

13. The nature of microsphere-microsphere adhesion remains unclear. Why does a thicker, more relaxed membrane promote adhesion? While the authors attempt an explanation, this fundamental aspect requires precise understanding supported by experimental data.

Response: We thank the reviewer for this important comment. We recognize that explaining microcapsule adhesion by membrane thickening is not appropriate. The primary mechanism underlying microcapsule–microcapsule adhesion should be attributed to the electrolyte-induced screening of electrostatic repulsion (ACS Nano 2022, 16, 21087–21097). Without electrolyte, CNF/GO and PDDA assemble into well-stratified structures via strong electrostatic interactions, with a predominantly positively charged outer surface and a negatively charged interior. This charge asymmetry leads to strong electrostatic repulsion between individual microcapsules, preventing

adhesion. Upon addition of electrolyte, increased ionic strength weakens the electrostatic complexation between CNF/GO and PDDA, thereby enhancing the mobility of components within the membrane and leading to the formation of a homogeneous membrane. This membrane contains a uniform distribution of positive and negative charges, enabling microcapsules to adhere upon contact. Meanwhile, the weakened electrostatic interactions also promote interdiffusion between CNF/GO and PDDA during complexation, which can lead to membrane thickening.

To further clarify the role of ionic strength in microcapsule adhesion, we prepared CNF–GO/PDDA prototissues at different NaCl concentrations. As shown in Fig. R19, at low NaCl concentrations (0.1 M), relatively strong electrostatic repulsion prevented microcapsule adhesion and prototissue formation. Stable prototissues were formed at intermediate NaCl concentrations (0.2 and 0.3 M), where effective microcapsule adhesion occurred. At a higher NaCl concentration (0.4 and 0.5 M), electrostatic interactions between CNF/GO and PDDA were excessively weakened, leading to extensive interdiffusion between the CNF/GO-containing extruded aqueous phase and the PDDA-containing collection bath, thereby inhibiting the formation of stable complex membranes and microcapsules.

We have added the above discussion in the revised manuscript. Fig. R19 has been included in the revised Supplementary Information as Supplementary Fig. 30.

Fig. R19. Influence of ionic strength on the construction of CNF-GO/PDDA prototissues by adding NaCl at 0.1 M (a), 0.2 M (b), 0.3 M (c), 0.4 M (d), and 0.5 M (e) into the extrusion droplets. Scale bar, 1 cm (left), 0.2 cm (right).

Characterization and Applications

14. The authors demonstrate various possibilities that have already been reported in the literature using their new prototissue types but provide insufficient characterization. Detailed mechanical property analysis of protocells and prototissues is necessary to support claims about mechanical properties and bending mechanics of layered prototissues. What forces are exerted during bending? **Response:** We appreciate the comment from the reviewer. The layered prototissue consists of two stacked layers: a CNF/PDDA prototissue and a CNF-GO/PDDA prototissue. To quantitatively compare the mechanical strength of CNF/PDDA and CNF-GO/PDDA prototissues, we separately prepared conical CNF/PDDA and CNF-GO/PDDA prototissues and immersed them in a 1 wt% PDDA-100k aqueous solution. The time-dependent volume shrinkage of each prototissue was recorded and quantified as the volume shrinkage ratio, calculated as $1 - V/V_0$ based on the cone

volume before and after shrinkage. As shown in Fig. R20, CNF/PDDA prototissue exhibits both a faster and larger volume reduction than CNF–GO/PDDA prototissue. After 1 hour, the volume shrinkage ratios of the CNF/PDDA and CNF–GO/PDDA prototissues reach 77% and 56%, respectively. These results indicate that CNF/PDDA prototissues possess lower mechanical strength and are more susceptible to osmotic-pressure-induced shrinkage.

Bending of the layered prototissue is not driven by an external mechanical load but arises from internal stress generated by asymmetric osmotic shrinkage between the two layers. Owing to their different mechanical strength and osmotic responsiveness, the CNF/PDDA and CNF–GO/PDDA prototissues undergo unequal magnitudes and rates of volumetric contraction under the same hypertonic conditions. This mismatch produces a through-thickness strain gradient, which drives bending of the bilayer structure.

We have added a brief discussion in the revised manuscript. Fig. R20 has been included in the revised Supplementary Information as Supplementary Fig. 34, along with a brief discussion.

Fig. R20. a, Side-view images of a CNF/PDDA prototissue before (left) and after 60 min immersion in 1 wt% PDDA-100k solution (right). b, Side-view images of a CNF–GO/PDDA prototissue before (left) and after 60 min immersion in 1 wt% PDDA-100k solution (right). c, Time evolution of the prototissue shrinkage for the two compositions. Error bars represent the standard deviation ($n = 3$). Scale bars, 5 mm.

15. For the buoyant prototissue: what is the intended purpose and biological relevance? What is the buoyancy rate and its dependence on experimental parameters?

Response: We appreciate the comment from the reviewer. The purpose of the buoyant prototissue

is to demonstrate that our DIC-assembled, large-scale prototissues can be endowed with enzyme-powered, buoyancy-controlled actuation. In this way, they can achieve vertical displacement, gas-driven shape editing, and cargo manipulation. Biologically, this design provides a minimal tissue-level analogue of gas-filled, buoyancy-regulating and shape-modulating structures in living systems e.g., gas vesicles in microorganisms or swim bladders in fish (*Nat. Rev. Microbiol.* **2008**, *6*, 466–476; *Nat. Rev. Microbiol.* **2012**, *10*, 705–715), and also illustrates how multicompartiment prototissues can function as programmable, enzyme-driven actuator modules that combine buoyancy control with reconfigurable morphology for potential applications in dynamic, bioinspired materials (*Small* **2024**, *25*, 2308580; *Angew. Chem. Int. Ed.* **2024**, *6*, e202311556).

In the revised manuscript we now quantify the buoyancy rate and its dependence on experimental parameters (Fig. R20). Here, the buoyancy rate is defined as the vertical ascent velocity of the prototissue, obtained from the slope of the normalized distance-time curves during the approximately linear rising stage after the induction period. As shown in Fig. R20 b,c, when the catalase content in the CNF–GO/PDDA prototissue is fixed at 0.1 wt%, increasing the H₂O₂ concentration from 3 wt% to 6 wt% accelerates bubble generation and leads to a higher ascent velocity, whereas further increasing the H₂O₂ concentration slows down the motion because of partial inhibition of catalase at excessive substrate levels. As shown in Fig. R20 d,e, when the external H₂O₂ concentration is fixed at 3 wt%, reducing the amount of encapsulated catalase leads to a systematic decrease in the buoyancy rate, thus the prototissue rises more slowly.

These data demonstrate that the buoyancy-driven actuation of the prototissue can be quantitatively tuned by both the external fuel concentration and the internal enzyme loading. Fig. R21 has been included in the revised Supplementary Information as Supplementary Fig. 35, along with a brief discussion.

Fig. R21. a, Digital images showing a catalase-loaded CNF-GO/PDDA prototissue rising in aqueous H₂O₂ after an induction period. Scale bar, 1 cm. b, Height-time traces for prototissues with fixed catalase content (0.1 wt%) in H₂O₂ solutions of different concentrations (3, 6 and 9 wt%). c, Corresponding buoyancy rates, defined as the vertical ascent velocity obtained from the slope of the approximately linear rising segment of the curves in b. d, Normalized distance-time traces for prototissues with different catalase loadings at a fixed H₂O₂ concentration (3 wt%). e, Buoyancy rates as a function of catalase loading. Error bars represent the standard deviation (n = 3).

Biocompatibility Claims

16. The authors claim biocompatibility in their conclusions without providing supporting data.

Triton-X usage could be problematic in this regard. While an in vitro cytotoxicity assay is mentioned in the experimental section, no data are presented in the main text.

Response: We appreciate the comment from the reviewer. For the biocompatibility of microcapsules, we performed an in vitro CCK-8 assay. As shown in Fig. R22, L929 cells were incubated with extracts from CNF-GO/PDDA microcapsules for 72 h, and cell viability was assessed by the CCK-8 assay. Over the tested concentration range, cell viability remained above 95%, indicating excellent biocompatibility. Regarding the reviewer's concern about the biocompatibility of Triton X-100, it can be readily replaced with biocompatible surfactants. As noted in our response to Comment 3, in addition to Triton X-100, microcapsules can also be successfully prepared using Pluronic F68, Pluronic F127, Tween-20, and Solutol HS-15. CNF-GO/PDDA microcapsules were also prepared using Pluronic F127 and Pluronic F68, and cytotoxicity assays likewise demonstrated that these CNF-GO/PDDA microcapsules exhibit excellent biocompatibility.

We have added the corresponding results to the revised Supplementary Information as Supplementary Fig. 17 with a brief discussion.

Fig. R22. **a**, Live/dead staining images of L929 cells after incubation with the corresponding material extracts (Control, Triton X-100, Pluronic F127, and Pluronic F68); live cells are stained green (Calcein-AM) and dead cells are stained red (PI), with merged images shown in the right column. Scale bar, 100 μm . **b**, CCK-8 assay results (OD value at 450 nm) of L929 cells cultured with the corresponding extracts for 1–3 days. Error bars represent the standard deviation ($n = 3$), and P values are indicated. [Surfactant] = 10^{-3} wt%.

Responses to Reviewer #3

Reviewer #3 (Remarks to the Author):

The authors describe the construction of prototissues via adhesive interaction between microgel compartments fabricated from PDDA and CNF. The membrane permeability of the compartments were studied and modulated by inclusion of GO. They also illustrated how osmotic changes can help regulate the morphology of the microgel compartments irreversibly via cross linking of the interior or reversibly via delayed permeability. They further went on to study a two-enzyme cascade within the CNF-GO/PDDA microcapsules and show chemomechanical actuation of the prototissues mediated by osmotic changes and bubble mediated buoyancy.

1) The primary concern is regarding novelty – as a significant part of the paper mostly deals with preparation of the protocells and their osmotic response along with enzyme cascade studies. These are routine studies in protocell field and the authors do well to thoroughly characterize their new protocell model/chasis but these do not add significant novelty to the work.

Response: We appreciate the reviewer's comment. In this work, we employed a new strategy to construct protocells using a new system. We agree with the reviewer that osmotic response and enzymatic reactions are well-established approaches in protocell research; however, owing to the fundamentally different nature of our system, a thorough understanding of these newly developed protocells is essential for subsequent prototissue construction and functional implementation. Beyond the novelty of the production method, our work addresses several key challenges in this field, including the large-scale preparation of protocell/prototissue in aqueous media, achieving excellent mechanical strength and stability, and enabling precise shaping of prototissues. In the revised manuscript, we have added substantial experimental data and discussion, addressing all reviewer questions and including additional results such as the periodic motion of prototissues, the preparation of gripper-like prototissues, and their ability to manipulate macroscopic objects, which further strengthen the significance of our work. We hope that the revised manuscript meets the reviewer's high expectations.

To further clarify the novelty of our contribution, we summarize below several bottlenecks and challenges identified in review articles by multiple groups.

Prof. Gobbo et al. (*Biochem. Soc. Trans.* **2020**, *48*, 2579–2589):

- “To date, prototissues have been assembled in the millimetre and submillimetre size range using a variety of techniques...A materials chemistry approach would allow the advancement of current prototissue construction technologies and enable the precise assembly of protocells into large-scale materials (>1 cm in size) that are robust, free-standing, and stable in water media. This is critical for future technological applications of prototissues and would represent a major achievement in the field.”
- “To date, research has been focusing on the development of methodologies to assemble protocells together in a controlled and reproducible manner. However, if we want to develop prototissue-based technologies, it is of paramount importance that we start advancing the prototissue’s mechanical properties. In this regard, current studies have been limited to the characterisation of the prototissue’s tensile strength and elastic modulus...It is now critical to construct protocellular materials with a range of mechanical properties that mimic those of living tissues and perhaps even outperform them.”

Prof. Han et al. (*Adv. Funct. Mater.* **2024**, *34*, 2405823):

- “Although prototissues with precise spatial structures can be assembled with micrometer and millimeter ranges, they are still small compared to tissues for tissue transplantation. To assemble large-scale prototissues, the long-distance signaling, synergistic effects between artificial cells, and collective functions will face enormous challenges.”

Prof. Lin et al. (*Adv. Healthcare Mater.* **2025**, *14*, 2500376):

- “Despite significant progress in the development of prototissues, several challenges remain... 2) Natural tissues exhibit a broad range of mechanical strength due to the ECM composition (e.g., collagen, elastin) and cellular interactions. The mechanical stability of prototissues or protocell assemblies is typically low; as such, the incorporation of synthetic hydrogels, scaffolds, or artificial ECM needs to be considered for structural support ...5) Engineering protocell assemblies into large-scale, functional 3D structures with batch-to-batch reproducibility remains challenging. Current fabrication techniques, including microfluidics and 3D printing, struggle to precisely control spatial organization at a tissue-like scale.”

2) The advantage of a prototissue is that it has compartments which can host a reaction to induce morphological or functional output. In the current design, the authors have used the osmotic

response of the protocells and bubble mediated buoyancy to produce actuation. The osmotic pressure mediated actuation due to differential response in the two layer tissue was also illustrated in earlier studies (G. Villar et al. Science 2013, 340, 48-52). The buoyancy-based actuation of prototissues which the authors have shown is quite interesting as it is mediated by reactions within the protocells. However, this reviewer would like to see not just actuation but also recovery and re-actuation over multiple cycles which would then constitute a significant advance in the field.

3) One suggestion would be to use Glucose oxidase mediated consumption of the oxygen bubble to execute multiple cycles of actuation or formulate a different strategy for dissolution of the bubble for recovery of the actuated tissue.

Response: We appreciate the reviewer's constructive suggestions. We achieved this goal by employing glucose oxidase (GOx)-mediated consumption of oxygen bubbles. As shown in Fig. R23, a small CNF-GO/PDDA prototissue encapsulating 0.1 wt% catalase and 0.1 wt% GOx was prepared and placed in a water-filled glass tube (diameter = 2 cm). An agarose hydrogel containing 3 wt% H₂O₂ was fixed at the bottom of the tube, while an agarose hydrogel containing 18 wt% glucose was fixed at the top. H₂O₂ and glucose gradually diffused from the hydrogels into surrounding water. When the prototissue approached the bottom H₂O₂ reservoir, catalase decomposed H₂O₂ into O₂, generating gas bubbles inside the protocells and increasing buoyancy, which caused the prototissue to rise. Upon reaching the top glucose reservoir, GOx consumed the encapsulated O₂, leading to a decrease in buoyancy and causing the prototissue to sink back toward the bottom. In this manner, the prototissue underwent repeated up-down motion between the two agarose hydrogels until the H₂O₂ and glucose were largely depleted, after which the prototissue gradually ceased motion and settled at the bottom. Under experimental conditions used, the prototissue exhibited three reproducible oscillation cycles.

We have added a discussion in the revised manuscript, and Fig. R23 is included as Fig. 7. 1-m. A supplementary movie has also been provided as Supplementary Movie 17.

Fig. R23. Schematic illustration (a) and digital images (b) showing the vertical oscillatory motion and displacement-time trace (c) of CNF-GO/PDDA prototissue. Scale bar, 1 cm.

4) The authors have studied the shrinkage of the CNF-GO/PDDA protocells with PEI and PDDA. It is seen that PDDA 100 kDa permeates and cross links but does not form a yolk structure but PEI forms a yolk structure which constitutes a highly cross linked network. Author could conduct FRAP studies to understand the condensation mediated changes in fluidity of the interior and also clarify the why there is difference in morphological outcomes for PEI and PDDA.

Response: We thank the reviewer for this important suggestion and have performed fluorescence recovery after photobleaching (FRAP) experiments using FITC-labelled PEI-0.6k and FITC-labelled PDDA-100k as fluorescent probes. As shown in Fig. R24, almost no fluorescence recovery was observed after photobleaching, indicating that both PDDA and PEI exhibit limited mobility. These results also indicate that the yolk structure inside the microcapsules formed by PEI/CNF/GO are solid-like aggregates rather than liquid-like coacervates (We note that after treatment with PDDA-100k, the microcapsules irreversibly collapse into crumpled shells. During confocal imaging, only the folded shell edges intersect the focal plane, resulting in a bright fluorescent rim and an apparently dark center that arises from out-of-focus effects rather than the absence of PDDA).

The distinct morphological outcomes observed for PEI-0.6k and PDDA-100k arise from two primary factors. First, the permanently charged quaternary ammonium groups on PDDA-100k interact more strongly with negatively charged CNF/GO than the protonatable amine groups on PEI-0.6k, leading to a more stable and robust electrostatically crosslinked network. Second, the much higher molecular weight and longer chain length of PDDA-100k allow electrostatic crosslinking with CNF/GO over a larger internal volume of the microcapsules, favoring the formation of a more extended and continuous network. Consequently, when shrunken CNF-GO/PDDA protocells

formed in PDDA-100k or PEI-0.6k solutions are redispersed in pure water, the stronger PDDA-100k/CNF-GO network effectively suppresses reswelling. In contrast, the weaker and more localized network formed with PEI-0.6k cannot withstand osmotic swelling, resulting in network disruption and the formation of small fragments that gradually associate into larger aggregates.

We have added a discussion in the revised manuscript, and Fig. R24c,d is included as Supplementary Fig. 24.

Fig. R24. a, Confocal fluorescence images of a CNF-GO/PDDA protocell containing PDDA-FITC before bleaching (-1 s), immediately after bleaching (0 s) and after 1200 s recovery, with the bleached region indicated by the arrow. Scale bar, 200 μm . b, The corresponding fluorescence intensity in the bleached region. c, Confocal fluorescence images of a CNF-GO/PDDA protocell containing PEI-FITC before bleaching (-1 s), immediately after bleaching (0 s) and after 1200 s recovery, with the bleached region indicated by the arrow. Scale bar, 200 μm . d, The corresponding fluorescence intensity in the bleached region.

5) the facile construction of the prototissue due to sticking interactions at high electrolyte concentrations has the limitation of lacking programmability of the internal structure of the prototissue – in terms of self-organization of different protocells to form complex hierarchical structures. This reviewer would like to see the potential of this prototissue design in terms of soft robotics, where these prototissues can manipulate macroscopic objects or these actuatable tissues can be used as artificial muscle to operate a gripper like structure. These experiments would add significant novelty to the manuscript.

Response: We appreciate the reviewer's comment. In the original manuscript, we have already

demonstrated two approaches to program the internal structure and functionality of prototissues: (1) by alternately stacking CNF–GO/PDDA and CNF/PDDA protocells, a Janus-type starfish-shaped prototissue was fabricated, which exhibited bending behavior under osmotic stimulation; (2) by integrating catalase-containing CNF–GO/PDDA protocells with catalase-free CNF–GO/PDDA protocells, different buoyancy-driven motion modes of prototissues were achieved.

Following the reviewer’s constructive suggestion, we designed and fabricated a soft gripper prototissue. As shown in Fig. R25, a cross-shaped CNF–GO/PDDA prototissue was first constructed. CNF–GO/PDDA protocells containing NdFeB ferromagnetic microparticles were then deposited in the central region of the cross, while CNF–GO/PDDA protocells containing catalase were deposited at the ends of the four arms. Upon addition of H₂O₂ to the aqueous environment, oxygen was generated inside the catalase-containing protocells, producing buoyancy that induced upward bending of the four arms and enabled the gripper to capture three hollow plastic spheres. Furthermore, directional movement and rotation of the loaded soft gripper could be achieved by applying an external magnetic field.

We have added a discussion in the revised manuscript, and Fig. R25 is included as Fig. 7j,k. A supplementary movie has also been provided as Supplementary Movie 16.

Fig. R25. a,b, Schematic illustration (a) and digital images (b) showing the cross-shaped prototissue functioning as a soft gripper. Scale bar, 2 cm.

6) Another general comment would be that it may be investigated what is the advantage of using a prototissue over an actuatable gel for carrying out these tasks.

Response: We thank the reviewer for this important comment. The choice of a protocell-based prototissue over a monolithic actuatable hydrogel is motivated by several complementary advantages in transport, spatial programmability and architecture, which are directly relevant to the

tasks demonstrated in our work.

First, in a bulk hydrogel actuator the response time is typically limited by diffusive transport of water and solutes through a continuous polymer network, so that the characteristic time increases strongly with size and can reach tens of minutes to hours for centimetre-scale gels (*Mater. Today* **2014**, *17*, 494–503). In our DIC-assembled prototissues, the material is instead built from many thin-shelled, highly permeable CNF/PDDA protocells. Fuel molecules and osmolytes only need to diffuse short distances to reach the active regions. As a result, even prototissues with characteristic dimensions up to ~10 cm can undergo pronounced osmotic reshaping and enzyme-powered, buoyancy-mediated motion on the timescale of a few minutes in most cases, whereas an equally large homogeneous gel would respond much more slowly under comparable conditions.

Second, achieving spatially programmed, multi-module behaviour in a single continuous hydrogel usually requires sophisticated patterning technologies such as multimaterial 3D printing or multi-step photo-patterning of orthogonal chemistries, which demand specialized equipment and elaborate formulations (*Matter* **2023**, *7*, 2419-2438; *Front. Robot. AI* **2021**, *8*, 673533). By contrast, the internal architecture of our prototissues is encoded simply by mixing and placing different protocell populations (e.g., catalase-loaded, magnetic, passive) at defined locations during assembly, without changing the base chemistry or fabrication setup.

Third, each protocell acts as a separate but communicating reaction compartment. This allows different microenvironments and reaction networks to coexist within a single prototissue, while the permeable shells support controlled exchange of substances with the external medium. Such a multicompartment design is more closely analogous to the hierarchical organisation of living tissues than a monolithic hydrogel (*Cell Biomater.* **2025**, *9*, 100097).

Manuscript ID: NCOMMS-25-49805B

Title: One-Step Construction of Robust Protocells and Prototissues in Water

Authors: Weixiao Feng, Peifan Li, Xin Li, Ziwei Wang, Min Chen, Yang Hu, Fu-Jian Xu*, Shaowei Shi*

Dear editor:

We would like to thank the reviewers for their detailed feedback to improve our submission. We have considered each comment carefully and revised our manuscript to address the issues raised.

Responses to Reviewer #1

Reviewer #1 (Remarks to the Author):

The questions have been well addressed and it is recommended for publication.

[Editorial note: Reviewer #1 addressed authors' responses to the concerns of Reviewer #2 and finds them adequately and satisfactorily addressed.]

Response: We appreciate the reviewer for the positive evaluation and for recognizing our efforts in addressing the concerns.

Responses to Reviewer #3

Reviewer #3 (Remarks to the Author):

The authors have significantly addressed many of the points raised by this reviewer which has increased the scope of the work. One key distinguishing point noted in the revised manuscript is the mechanical strength of the fabricated prototissues which is quite commendable. **Response:** We appreciate the reviewer for the positive evaluation and for recognizing our efforts in addressing the concerns.

Minor points of concern remain which are discussed below:

1) The key focus of the manuscript is fabrication of prototissues and in order to mimic prototissues it is essential to have a membrane permeability cut-off which can encapsulate enzymes. The higher permeability of the CNF-GO/PDDA microcapsules fabricated can only entrap small enzymes such as Horse Radish Peroxidase (44 kDa) as also seen in their data where the peroxidase is only found entrapped in the membrane not in the lumen. The permeability of the microcapsules has been adjusted by adding the GO – can this be taken forward more to get more stringent molecular weight cut-off to be able to encapsulate enzymes? If yes, add relevant data or these limitations and potential ways to overcome this may be discussed in the conclusion.

Response: We thank the reviewer for this comment. We note that the apparent enrichment of

horseradish peroxidase (HRP) in the membrane mainly arises from its participation in diffusion-inhibited complexation (DIC) via electrostatic interactions during microcapsule formation, rather than from the high permeability of CNF–GO/PDDA microcapsules. We have clarified this in the revised manuscript as follows: “Due to the negative charge of both HRP and GOx under neutral conditions, they participated in membrane formation via DIC, as evidenced by the strong fluorescence signals of FITC-labelled HRP and RITC-labelled GOx on the membrane. HRP and GOx were also distributed inside the protocells, with a lower content of HRP detected. Compared to GOx, HRP contained more hydrophobic groups, enhancing its interfacial activity at the air–water interface and facilitating the DIC process, similar to the case of GO-containing protocell.”

We fully agree with the reviewer that tuning membrane composition (e.g., adjusting GO content) could further regulate permeability and enable a more stringent molecular-weight cut-off for broader enzyme encapsulation. However, such tuning may affect not only permeability but also mechanical strength and structural stability of microcapsules, and requires systematic investigation. We therefore added a brief statement in the conclusion: “Despite these advances, several aspects remain to be explored. For example, further tuning of membrane composition and/or post-modification of the membrane could enable more precise control over permeability and a tighter molecular-weight cut-off, which are important for broadening the range of encapsulation.”

2) The authors have fabricated a soft gripper powered by buoyant forces. The limitation of such design is that the forces are only acting along the vertical axis. The osmotic actuation is not enzyme driven and hence not as interesting, but the buoyant design seems limited in its outlook. These limitations may be discussed and future plan for improvements may be discussed in the conclusion.

Response: We thank the reviewer for this comment. As described in the revised manuscript (Figure 7j,k and Supplementary Video 16), in addition to buoyancy- and osmosis-driven actuation, we also demonstrated magnetic-field actuation: “By further incorporating protocells containing NdFeB ferromagnetic microparticles into the central region, the resulting soft gripper could be directed to translate and rotate under an external magnetic field.” Accordingly, integrating additional actuation mechanisms (e.g., light and temperature) is expected to enable more complex and programmable motion and deformation of prototissues. We have added this outlook in the conclusion.